# Synthesis of New Biscoumarin Derivatives, In Vitro Cholinesterase Inhibition, Molecular Modelling and Antiproliferative Effect in A549 Human Lung Carcinoma Cells

**DOI:** 10.3390/ijms22083830

**Published:** 2021-04-07

**Authors:** Monika Hudáčová, Slávka Hamuľaková, Eva Konkoľová, Rastislav Jendželovský, Jana Vargová, Juraj Ševc, Peter Fedoročko, Ondrej Soukup, Jana Janočková, Veronika Ihnatova, Tomáš Kučera, Petr Bzonek, Nikola Novakova, Daniel Jun, Lucie Junova, Jan Korábečný, Kamil Kuča, Mária Kožurková

**Affiliations:** 1Department of Biochemistry, Faculty of Science, Pavol Jozef Šafárik University in Košice, Šrobárova 2, 041 54 Košice, Slovakia; monika.hudacova@student.upjs.sk (M.H.); eva.konkolova@uochb.cas.cz (E.K.); maria.kozurkova@upjs.sk (M.K.); 2Department of Organic Chemistry, Institute of Chemical Sciences, Faculty of Science, Pavol Jozef Šafárik University in Košice, Moyzesova 11, 040 01 Kosice, Slovakia; 3Institute of Organic Chemistry and Biochemistry of the Czech Academy of Sciences, Flemingovo Náměstí 542/2, 160 00 Prague 6, Czech Republic; 4Department of Cellular Biology, Faculty of Science, Pavol Jozef Šafárik University in Košice, Šrobárova 2, 041 54 Košice, Slovakia; rastislav.jendzelovsky@upjs.sk (R.J.); jana.vargova@upjs.sk (J.V.); juraj.sevc@upjs.sk (J.Š.); peter.fedorocko@upjs.sk (P.F.); 5Department of Toxicology and Military Pharmacy, Faculty of Military Health Sciences, University of Defense, Trebesska 1575, 500 05 Hradec Kralove, Czech Republic; ondrej.soukup@fnhk.cz (O.S.); veronika.ihn@gmail.com (V.I.); tomas.kucera2@unob.cz (T.K.); bzonepe@gmail.com (P.B.); daniel.jun@unob.cz (D.J.); lucie.junova@unob.cz (L.J.); korabecny.jan1@gmail.com (J.K.); 6Biomedical Research Centre, University Hospital Hradec Kralove, Sokolska 581, 500 05 Hradec Kralove, Czech Republic; jana.janockova@upjs.sk (J.J.); nikola.novakova.2@uhk.cz (N.N.); kamil.kuca@uhk.cz (K.K.); 7National Institute of Mental Health, Topolová 748, 250 67 Klecany, Czech Republic

**Keywords:** biscoumarin, Alzheimer’s disease, blood–brain barrier, cholinesterase, antiproliferative activity, A549

## Abstract

A series of novel C4-C7-tethered biscoumarin derivatives (**12a**–**e**) linked through piperazine moiety was designed, synthesized, and evaluated biological/therapeutic potential. Biscoumarin **12d** was found to be the most effective inhibitor of both acetylcholinesterase (AChE, IC_50_ = 6.30 µM) and butyrylcholinesterase (BChE, IC_50_ = 49 µM). Detailed molecular modelling studies compared the accommodation of ensaculin (well-established coumarin derivative tested in phase I of clinical trials) and **12d** in the human recombinant AChE (*h*AChE) active site. The ability of novel compounds to cross the blood–brain barrier (BBB) was predicted with a positive outcome for compound **12e**. The antiproliferative effects of newly synthesized biscoumarin derivatives were tested in vitro on human lung carcinoma cell line (A549) and normal colon fibroblast cell line (CCD-18Co). The effect of derivatives on cell proliferation was evaluated by MTT assay, quantification of cell numbers and viability, colony-forming assay, analysis of cell cycle distribution and mitotic activity. Intracellular localization of used derivatives in A549 cells was confirmed by confocal microscopy. Derivatives **12d** and **12e** showed significant antiproliferative activity in A549 cancer cells without a significant effect on normal CCD-18Co cells. The inhibition of *h*AChE/human recombinant BChE (*h*BChE), the antiproliferative activity on cancer cells, and the ability to cross the BBB suggest the high potential of biscoumarin derivatives. Beside the treatment of cancer, **12e** might be applicable against disorders such as schizophrenia, and **12d** could serve future development as therapeutic agents in the prevention and/or treatment of Alzheimer’s disease.

## 1. Introduction

Coumarins are either natural or synthetic analogues of oxygen-containing heterocycles of a typical benzopyrone structure with a rich electron count and good charge transport properties [1]. The coumarin backbone is used extensively in the synthesis of a wide range of functional molecules for biological diagnosis and also as probes that display a variety range of biological activities, such as anticoagulant, antineurodegenerative, antioxidant, anti-inflammatory, antidiabetic, antidepression, anti-cancer, and antimicrobial effects [2,3,4,5,6]. Specifically, 4-hydroxycoumarin derivatives have attracted considerable interest in several fields of research [7,8,9].

Biscoumarins, an important class of molecules, emerged as privileged natural scaffolds with various biological properties [7,8,9,10]. Biscoumarins can interact with biomacromolecules, a feature that is extensively pursued in anti-cancer (Figure 1) [11,12,13,14,15] and anti-Alzheimer therapy (Figure 2) [16,17,18]. Although both types of diseases are multifactorial primary diseases and are related to the aging process, they affect millions of people worldwide. Thus, both diseases represent a growing burden for public health systems, which makes this topic relevant [19,20].

While Alzheimer’s disease (AD) is characterized by massive neurodegeneration and loss of tissue in the brain, cancer is characterized by a substantial increase in cell number due to uncontrolled cell division [21]. Thus, the pathogenesis of AD and cancer have an inverse association. However, many recent studies indicate the existence of multiple signaling pathways in tumorigenesis that are associated with neurodegenerative disease and may contribute to the onset and progression of both disorders [20,21]. Over the last few years, several studies have examined several different anti-cancer drugs to investigative their coactivity under neuropathological conditions including AD [22]. Anti-cancer drugs with moderate to strong blood–brain barrier (BBB) permeability scores may act as anti-amyloidogenic and brain penetrant microtubule-stabilizing agents or acetylcholinesterase (AChE) inhibitors (AChEIs). The dual binding site AChEIs can bind to catalytic anionic sites (CAS) along with peripheral anionic sites (PAS) and inhibit enzymatic activity AChE and AChE-inducted Aβ peptide aggregation [23,24]. Several studies have shown that the coumarin moiety can interact with the PAS of AChE and inhibit the activity of the enzyme and also act as antioxidant and chelating agents [16,17,24,25,26,27].

Lung cancer is among the most common types of cancer worldwide, with average of two million new cases every year [28]. Despite innovations in diagnostic testing, surgical techniques, and chemotherapeutic treatments, the five-year survival rate of lung cancer patients remains low (13–15%) [29]. These unfavorable statistics forced pharmacological research to look more deeply into the lung cancer molecular mechanism and, second, to develop new therapies for improving patient survival and quality of life.

As already mentioned, the AChE is an enzyme primarily expressed in the central and peripheral nervous systems, although different isoforms of AChE are also constitutive of various cell types and tissues, such as lung. Abnormal expression and structural alteration of AChE and multiple activities have been observed in different types of tumors, such as brain, lung, ovarian, breast, hepatocellular, renal, and colon cancer [29,30,31]. The main role of AChE is catalytic hydrolysis of cholinergic neurotransmitters [32]. However, recent studies have shown the non-classical function of the enzyme as a potential tumor suppressor and regulator of apoptosis, which support the involvement of AChE in the regulation of tumor development. The non-classical function AChE is based on the ability of AChE to bind with a range of proteins through the PAS [33], which is located near the entrance of catalytic gorge. This raises the possibility that other proteins may hinder the access of the substrate to the catalytic triad of AChE [34]. In lung cancer, the AChE activity is reduced, and the level of acetylcholine (ACh) is increased. The ACh was demonstrated to be an autocrine and paracrine growth factor for lung epithelial cells [35,36,37].

The regulation of AChE activity by small molecules has been investigated as a strategy for lung cancer therapy. The simple assumption is that small molecules, which increase activity of AChE in human lung cancer cells, will reduce the levels of ACh. Indeed, it was demonstrated that the increased activity level of AChE suppresses proliferation, growth, and survival of human lung cancer [38]. Accordingly, the decrease in the functional activity of AChE and butyrylcholinesterase (BChE) by using their inhibitor promoted the proliferation of human lung cancer cells by maximizing/increasing the level of the growth factor ACh using their inhibitor in human cancer tissue [30,39]. Interestingly several AChEI have been investigated for their growth-inhibitory activity in human lung cancer cell line, and some of them truly suppressed colony formation of cancer cells in dose-dependent manner [40,41,42,43,44,45]. This report suggested that compounds could have many pleotropic biological effects apart from suppressing AChE activity. Thus, molecular mechanisms of action of the small molecules on non-catalytic function of AChE in lung and other cancer associated with increased level of AChE are the interesting subject of further studies.

Moreover, previous studies indicate that modulation of AChE via AChEIs can be used as a possible anticancer strategy. These various biological effects only confirm that there is a very fine line between the AChEIs via modulating AChE activity that has acted as a suppressor of cancer cell proliferation or conversely as an initiator of cancer cell proliferation. On other hand, it can be seen from previous studies, the effect of especially catalytic AChEIs on cancer treatment is not favorable [30,39]. The identification of potential anticholinesterase activity of anticancer designed drugs, in the primary study, may be good parameter for determining if the drug could be used for cancer treatment or not [46], and vice versa, drugs for AD therapy are required to have low toxicity against cells in general [47,48]. It was previously mentioned that pharmacotherapies have been still sought as new potential molecules in the treatment of cancer or AD, which belong to the most prevalence diseases in the world [19,20].

Given that coumarin derivatives are well known for their biological activity and are important building blocks for the synthesis of other biologically active compounds, we decided to synthesize novel biscoumarin derivatives and study their selected activities on cancer and normal cells. In this study, we report the design, synthesis, biological evaluation, and molecular modelling studies of a series of coumarin derivatives **12a**–**e**, based on a combination of two units of coumarin scaffold via piperazine-alkoxy side chain. The novel derivatives were then tested for their inhibition effect on human recombinant AChE (*h*AChE)/human recombinant BChE (*h*BChE), ability to cross the BBB, antiproliferative activity on human lung carcinoma cell line (A549), selectivity towards normal colon fibroblast cell line (CCD-18Co), and detailed molecular modelling studies. This study was done to determine whether new biscoumarine derivatives can be used to treat AD or cancer.

## 2. Results and Discussion

### 2.1. Chemistry

The target biscoumarin homodimers, derivatives **12a**–**e**, were synthesized according to the process shown in Figure 3. The coumarin derivatives were synthesized from resorcinol with citric acid in the presence of concentrated sulfuric acid [49]. The methylation of (7-hydroxy-2-oxo-2*H*-4-chromenyl) acetic acid produced the methyl ester, compound **9** [50]. The (7-hydroxy-2-oxo-2*H*-4-chromenyl) acetic acid methyl ester (**9**) reacted with 1-(2-aminoethyl)piperazine in acetonitrile to produce 2-(7-hydroxy-2-oxo-chromen-4-yl)-*N*-(2-piperazin-1-ylethyl)acetamide (**10**). The reaction of (7-hydroxy-2-oxo-2*H*-4-chromenyl) acetic acid methyl ester (**9**) with appropriate *α*,*ω*-dibromoalkane (*n* = 3, 4, 6–8) in acetone produced intermediates methyl 2-[7-(*ω*-bromoalkoxy)-2-oxo-chromen-4-yl]acetates (**11a**–**e**). These key intermediates, derivatives **11a**–**e**, underwent reactions with 2-(7-hydroxy-2-oxo-chromen-4-yl)-*N*-(2-piperazin-1-ylethyl)acetamide (**10**) in the presence of K_2_CO_3_ and KI in acetone to provide the target biscoumarin derivatives **12a**–**e**.

### 2.2. Biological Profil of Biscoumarine Derivatives as Potential Drugs for Treatment AD

#### 2.2.1. Evaluation of *h*AChE and *h*BChE Inhibitory Activity

The in vitro cholinesterase activity of the new series of biscoumarin derivatives, compounds **12a**–**e**, was determined using spectrophotometry with tacrine and 7-methoxytacrine (7-MEOTA) as reference compounds (Table 1). The obtained half maximal inhibitory concentration (IC_50_) values suggest that derivatives **12c** and **12d** exhibited potent and selective inhibitory activities at micromolar concentrations. Compounds **12a**, **12b**, and **12e** did not show inhibitory activity against either *h*AChE or *h*BChE. The most potent compound against *h*AChE from the series was **12d** (IC_50_ = 6.30 µM), outperforming 7-MEOTA. With the exception of **12d** showing only marginal activity, all the derivatives were completely devoid of *h*BChE inhibition (IC_50_ > 500 µM), suggesting their selectivity pattern for *h*AChE.

#### 2.2.2. Molecular Modelling Studies

To envisage the accommodation of the top-ranked *h*AChE inhibitor **12d** in the active site of the respective enzyme, molecular modeling studies were performed. The template structure *h*AChE complexed with donepezil was taken from the Protein Data Bank database (PDB ID: 4EY7) [52]. The rationale for this selection stems from the fact that the enzyme–ligand complex is solved at high resolution (2.35 Å), and that donepezil represents dual-binding ligand with simultaneous affinity to both PAS as well as CAS of *h*AChE. For comparative purposes, we also carried out a docking simulation of ensaculin (KA-672) into the cavity of *h*AChE. KA-672 is a small molecule containing coumarin scaffold that reached phase I in clinical trial testing. Besides well-pronounced AChE inhibition (rat brain AChE IC_50_ = 0.36 μM) [18], KA-672 is also characterized by the affinity towards NMDA, serotonergic (5-HT_1A_, 5-HT_7_), adrenergic (*α*1), and dopaminergic (D2, D3) receptors, all being responsible for the complex action of the drug [53].

From the docking experiments, it is apparent that KA-672 is a dual binding inhibitor of AChE (Figure 4A,B). The benzopyranone moiety is buried deep facing in parallel to Trp86 by *π–π* stacking at a distance of 3.6 Å. The keto group of KA-672 forms a hydrogen bond with hydroxyl of Tyr133 (2.4 Å). The methoxy group of benzopyranone moiety and oxygen from the linker are engaged in another hydrogen bond with a hydroxyl group of Tyr124 (2.3 and 1.9 Å, respectively). The protonated nitrogen of piperazine moiety established cation–π interaction to Tyr341 (3.5 Å). At the mouth of the enzyme, 2-methoxybenzene generated parallel *π–π* interaction (3.6 Å). The catalytic triad residues (Ser203, His447, Glu334) seem to be unaffected by the ligand accommodation.

Likewise, **12d** also revealed AChE dual-binding site character (Figure 4C,D). At the bottom of the gorge, a proximal coumarin ring is anchored to Trp86 by parallel *π–π* interaction (3.7 Å). The methylester group is oriented towards the oxyanion hole of the enzyme defined by glycine residues with additional hydrogen bond to hydroxyl from Ser203. The alkoxy chain is implicated in several hydrophobic interactions in the mid-gorge region of the enzyme with, e.g., Trp86, Tyr337, Tyr124, and Phe338. Congruently to the KA-672-*h*AChE complex, the piperazine moiety of **12d** forms cation–*π* contact to Tyr341 (5.0 Å). Tyr341 and Tyr72 are also engaged in distorted *π–π* interaction with the distal coumarin ring. More importantly, the 2*H*-chromen-2-one is stacked against Trp286 (3.7 Å) in the PAS of the enzyme. The hydroxyl group of the distal coumarin ring revealed a hydrogen bond with hydroxyl moiety of Tyr72.

Taking the aforementioned together, both docking poses of KA-672 and biscoumarin derivative **12d** predicted the ligand’s dual-binding site character in the gorge of *h*AChE. More rotatable bonds in **12d** than in KA-672 generate a higher entropic penalty, which is the plausible culprit for one order of magnitude decreased inhibition potency of **12d**.

#### 2.2.3. In Vitro BBB Permeation Assays

The ability of agents to cross the BBB is a limiting factor in central nervous system (CNS) drug discovery. The efficiency of the studied biscoumarin derivatives **12a**–**e** was predicted using a parallel artificial membrane permeability assay (PAMPA). PAMPA is commonly used as an in vitro model of passive transcellular permeation and determines the permeability of compounds from a donor compartment through a lipid-infused artificial membrane into an acceptor compartment [51]. The data obtained from the assay are listed in Table 1. Reference drugs with known BBB permeability were used as controls. If the tested derivatives displayed values of permeability coefficient (*Pe*) over 4.0 × 10^−6^ cm s^−1^, they could potentially cross the BBB. Among the tested homodimers, only derivative **12e** demonstrated a high probability of BBB permeability. Derivatives **12c** and **12d** fell within the interval of *Pe* 2.0–4.0 ×10^−6^ cm s^−1^, suggesting an uncertain BBB permeation by passive diffusion.

### 2.3. In Vitro Antiproliferative Activity and Intracellular Localization of Analyzed Compounds

#### 2.3.1. Determination of Metabolic Activity and IC_50_ Values

MTT ((3-[4,5-dimethylthiazole-2-yl]-2,5-diphenyltetrazolium bromide)) assay is a well-established colorimetric assay that can be used to detect the effect of agents on cellular metabolism. The assay is based on the cleavage of the yellow tetrazolium salt, MTT, to purple formazan crystal by mitochondrial dehydrogenases. For the reaction to occur, the cells must not only be alive, but must also be metabolically active [54]. The effect of biscoumarin derivatives **12a**–**e** on cellular metabolism was examined using the MTT assay. As depicted in Figure 5 (Appendix A), the A549 cells did not lose the ability to convert dye even after 24 and 48 h treatment with derivatives **12a**, **12b**, and **12c** at concentrations ranging from 10 to 100 μM. However, a significant decrease (*p* < 0.001) in metabolic activity of the cells was observed after treatment with derivatives **12d** and **12e** (Figure 5A(ii), (iii)). The activity of derivatives **12d** and **12e** was also tested against CCD-18Co fibroblasts (Figure 5A(j),(jj)). Both derivatives **12d** and **12e** were found to reduce metabolic activity of CDD-18Co cell line in a much lesser extent compared to A549 cancer cell line. Derivative **12d** significantly reduced metabolic activity only at a concentration of 100 μM after 24 (*p* < 0.05) or 48 h (*p* < 0.001) treatment. The derivative **12e** significantly reduced metabolic activity in a time- (*p* < 0.05) and concentration-dependent manner (*p* < 0.05, *p* < 0.01). The IC_50_ values were calculated for derivatives **12a**–**e** to investigate the effect of structural change on their activity, and the results are shown in Table 2.

The obtained data revealed that while derivatives **12a**–**c** displayed no activity against A549 cancer cells, derivative **12d** showed moderate activity, with IC_50_ values of 94 and 58 μM after 24 and 48 h, respectively. Among the tested compounds, derivative **12e** showed the highest effect on cellular metabolic activity, with IC_50_ values of 49 and 35 μM after 24 and 48 h, respectively.

Derivatives **12d** and **12e** showed similar activity against A549 cancer cells to that reported for a series of 7-hydroxycoumarin derivatives containing urea-piperazine group [55] and dimers of triphenylethylene-coumarin hybrid containing a single amino side chain [11]. In the study by Huang et al. [55], the effect of the 7-hydroxycoumarin derivatives was compared with that of the commercially available anti-cancer drugs 5-fluorouracil and cisplatin on A549 cancer cell line (5-fluorouracil, IC_50_ = 36.34 µM; cisplatin, IC_50_ = 13.48 µM) and a human umbilical vein endothelial cell (HUVEC) cell line (5-fluorouracil, IC_50_ = 25.12 µM; cisplatin, IC_50_ = 9.68 µM). The results showed that HUVEC cell lines were more resistant to the coumarin derivatives than the tested commercially available drugs [13,55]. The effect of biscoumarin derivatives compounds **12d** and **12e** against the CCD-18Co fibroblast cell line showed a similar trend, making them promising candidates for future in vivo studies for their potential selective anti-cancer properties.

The MTT assay firstly suggests that divergences in the IC_50_ of derivatives **12a**–**e** might be associated with differences in the chemical structure of the molecules, for example, in the number of carbons in the alkoxy linker. These types of discrepancies in the activities of derivatives with increasing side-chain lengths are quite common [56,57]. Theoretical logP and logD values of compounds were also computed. The logP values play a prominent role in determining the activity of compounds, while both logP and logD values are important factors in membrane permeability and, thus, in biological activity. The studied biscoumarin derivatives had logP values in the range of 1.27–3.57 and logD values in the range of 0.67–2.68 (Table 2). The increasing length of alkoxy linker of biscoumarin derivatives led to decreasing metabolic activity of A549 cells; this change might be interpreted as a result of the interplay of two factors: (i) an increase in hydrophobicity that would be beneficial to the optimized insertion of the molecules into the phospholipid bilayer of the cytoplasmic membrane of the cells, and (ii) long chains might interfere with the “flip-flop” activity (the change in orientation in the membrane) that would be essential for K^+^ transport (potassium ion homeostasis in the cell is also of importance) and, therefore, for the biological activity of the derivatives [58]. The estimated differences in the accumulation of the biscoumarin derivatives **12a** (with shortest linker, *n* = 3) and **12e** (with longest linker, *n* = 8) in A549 cells were examined by densitometric analysis (in Section 2.3.5).

#### 2.3.2. Quantification of Cell Number and Viability

A quantification of cell number and viability was investigated to determine whether the effect of derivatives **12a**–**e** is cytostatic or cytotoxic. Cytostatic compounds delay the growth of cells without causing cell death, while cytotoxic compounds can cause massive cell death [59].

A549 cell line treated with derivatives **12a**–**e** at concentrations of 35 and 65 μM for 24 h showed no significant reduction in cell viability (i.e., cytotoxic effect) in any of the tested samples except for **12e**. Indeed, **12e** reduced cell viability at a concentration of 65 μM (*p* < 0.05) (Figure 5B(i)). However, all of the studied derivatives induced a significant reduction in cell number at both concentrations (**12a**: *p* < 0.01; **12b**: *p* < 0.05, *p* < 0.001; **12d**: *p* < 0.01, *p* < 0.001; **12e**: *p* < 0.001). These results indicate that the effect of the biscoumarin derivatives on the A549 cell line is cytostatic (Figure 5B(ii)) except derivative **12e** that in a higher concentration (65 μM) has both the significant cytostatic (*p* < 0.001) and the cytotoxic (*p* < 0.05) effect.

#### 2.3.3. Cell Cycle Analysis

Flow cytometric analysis with propidium iodide (PI) staining was used to determine whether or not the identified compounds could affect cell cycle distribution. PI binds to the nuclei of dead cells through the permeabilized plasma membrane, and the fluorescence of PI can be subsequently measured [60]. The A549 cells were treated with derivatives **12a**–**e** at concentrations of 35 and 65 µM, and the changes in the cell cycle were measured 24 h after treatment.

As shown in Figure 6, the coumarin derivatives (**12c**–**e**) induced changes in cell cycle distribution at both tested concentrations. In particular, derivatives **12d** and **12e** induced an increase in the percentage of the cell population in the G0/G1 phase (*p* < 0.001). A modest, yet statistically significant, decrease in the cell population in the S (35 μM: *p* < 0.05, 65 μM: *p* < 0.001) and G2/M (*p* < 0.001) phases was also observed. The derivative **12c** increased the accumulation of cell population in the G0/G1 phase (65 μM *p* < 0.01), which was accompanied with decreased cell population in the S (65 μM *p* < 0.01) and G2/M (35 and 65 μM *p* < 0.05) phases.

#### 2.3.4. Evaluation of Clonogenic Survival

The colony-forming assay is a valuable test for evaluating the efficiency of new drugs that are intended to reduce the population of cancer cells after treatment. It determines the ability of affected cells to proliferate and form new colonies. The predictive assay is of particular interest in assessing promising candidates developed as new anti-cancer agents [61].

To investigate the ability of the new biscoumarin derivatives to restrict colony formation, the A549 cell line was treated with derivatives **12a**–**e** at concentrations of 35 and 65 μM for 24 h. The cells were then harvested, and 500 viable cells of each experimental group were seeded. The ability of affected cells to form new colonies was evaluated after seven days of cultivation, and the results are shown in Figure 5C. In comparison to the untreated control, only derivative **12e** at a concentration of 65 μM exerted a 20% reduction in a relative number of colonies (*p* < 0.01). For all other derivatives, a higher but insignificant colony-forming ability of cancer cells was observed (increase of approximately 25 and 20% at the concentrations of 35 and 65 μM, respectively).

Beside these results and assumptions, we have observed a difference in cytotoxic and cytostatic effect, cell cycle progression, and colony forming in the A549 cell line after 24 h treatment derivatives **12a**–**12e** at concentrations of 35 and 65 µM.

These findings are in line with the growth inhibition observed through cellularity measurements and the MTT assay, which suggests that these divergences in the activity of derivatives **12a**–**e** might be associated with increasing lipophilic character of the chemical structure. Whether the difference in proliferation, viability, clonogenic assay, or cell cycle arrest is via to AChE inhibition by derivatives **12c** and **12d** is not entirely clear, and this will, therefore, need to be examined in future study.

However, the results from the colony-forming assay were comparable to the results from cellularity and viability assays, where compound **12e** was the most cytotoxic against A549 cells and arrested cell cycle, too. Therefore, in subsequent studies, we analyzed whether the incubation of cells with **12e** affected the mitotic activity of cells and onset of apoptosis by immunolabeling and confocal microscopy.

#### 2.3.5. Intracellular Localization and Mitotic Activity of A549 Cells

To investigate the intracellular localization and effect of the studied biscoumarin derivatives on mitotic activity and induction of apoptosis, human lung A549 adenocarcinoma cell line was used as an in vitro model. In the experiment, no sign of altered morphology or cell density in cells incubated with any of the studied derivatives (**12a**–**e**) was observed in comparison to the control samples (cells incubated with 10% fetal bovine serum (FBS) or 10% FBS with 1% dimethyl sulfoxide (DMSO)). Under normal conditions, the A549 cells possess a multipolar/fibroblastic cell shape and very low levels of autofluorescence after excitation with a 405 nm light (Figure 7A). Similar results were obtained in samples incubated with derivative **12a**–**12d** at concentrations of 35 and 65 µM (Figure 7B). In contrast, a markedly increased emission upon exposure to a 405 nm light was observed in cell bodies in samples incubated with derivative **12e** at both concentrations (35 and 65 µM), indicating the accumulation of the fluorescent chemical derivative in the cells (Figure 7C). The signal was predominantly present in the cytoplasm, but it did not appear to be co-localized exclusively with the mitochondria. A slightly weaker signal was also present in the nuclei. Collectively, our data suggest that, in contrast to derivative **12a**–**12d**, derivative **12e** can penetrate the cells after co-cultivation and accumulates both in the cytoplasm and in the nuclei. The differences between the accumulation of derivatives **12a** and **12e** in cells were also confirmed with densitometric analysis (Figure 7D).

In the next step, we analyzed whether the incubation of cells with coumarins affected the ability of the cells to proliferate. According to our results, 90.6 ± 5.7% of the A549 cells were found to proliferate under normal conditions (Figure 7E,F). In the group incubated with derivative **12a**, no significant changes were observed (Figure 7E,G), while in the group incubated with derivative **12e**, the number of proliferating cells decreased significantly to 59.4 ± 2.6% (at a concentration of 35 µM) and even to 16.5 ± 12.9% (at a concentration of 65 µM) (Figure 7E,H). Based on the visualization of a large fragment (89 kDa) of cleaved PARP protein, which is known to serve as a reliable indicator of apoptosis, it was observed that apoptotic cells were extremely rare (≤0.001%) in both the control groups and in cells incubated with **12e** (e.g., Figure 7F,H). Our analyses confirmed that despite the antiproliferative effect of derivative **12e**, the cells did not tend to undergo programmed cell death.

## 3. Materials and Methods

### 3.1. Experimental Part

All reagents used in the synthesis were obtained commercially and were used without further purification, unless otherwise specified. The reactions were monitored using thin-layer chromatography (TLC) with TLC sheets ALUGRAM-SIL G/UV254 (Macherey Nagel, Germany). Purification by flash chromatography was performed using silica gel 60 Å (0.0040–0.063 mm, Merck, Darmstadt, Germany) with the indicated eluent. Melting points were determined using a Boetius hot-stage apparatus and are presented in uncorrected form. NMR spectra were recorded at room temperature (RT) on a Varian Mercury Plus (Varian) 400 MHz spectrometer operating at 400 MHz for ^1^H and 100 MHz for ^13^C and on a Varian VNMRS 600 MHz spectrometer (operating at 600 MHz for ^1^H and 150 MHz for ^13^C). Chemical shifts (*δ* in ppm) are given from the internal solvent, DMSO-*d*_6_.

### 3.2. Synthesis of Biscoumarin Derivatives ***12a**–**e***

A mixture of methyl 2-(7-(bromoalkoxy)-2-oxo-2*H*-chromen-4-yl)acetate **11a**–**e** (0.225 mmol), 7-hydroxy-4-[(2-piperazinoetyl)amino]-2*H*-2-chromenone (10, 0.337 mM), K_2_CO_3_ (0.158 mM) KI in acetone (10 mL) was refluxed for 27–31 h. The reaction mixture was allowed to cool to room temperature, and the precipitate was filtered. The crude product was purified using column chromatography with MeOH/CHCl_3_ (5:1) elution to produce the desired product.

### 3.3. Spectroscopic Data

Methyl 2-(7-{2-[4-(2-{[2-(7-hydroxy-2-oxo-2*H*-4-chromenyl)acetyl]amino}ethyl) piperazino]propoxy}-2-oxo-2*H*-4-chromenyl)acetate (**12a**): The crude product purified by column chromatography on silica gel using a mixture of methanol and chlorophorm yielded yellow solid in 30%. Mp. 95–98 °C. ^1^H NMR (400 MHz, CD_3_OD-*d*_4_, *δ* ppm): 1.99–2.04 (m, 2H, CH_2_, H-13′), 2.45–2.58 (m, 12H, 6 × CH_2_, H-5′, 7′, 8′, 10′–12′), 3.32 (s, 4H, 2 × CH_2_, H-1′, 1‴), 3.35 (dt, 2H, CH_2_, H-4′, J = 12.8; 6.0 Hz), 3.72 (s, 3H, OCH_3_, H-4‴), 4.11–4.14 (m, 2H, CH_2_, H-14′), 6.18 (s, 2H, 2 × CH, H-3, 3″), 6.66 (d, 1H, CH, H-8″, J = 2.4 Hz), 6.76 (dd, 1H, CH, H-6″, J = 2.4; 8.4 Hz), 6.92 (d, 1H, CH, H-8, J = 2.4 Hz), 6.94 (dd, 1H, CH, H-6, J = 2.4; 9.2 Hz), 7.57 (d, 1H, CH, H-5″, J = 9.2 Hz), 7.61 (d, 1H, CH, H-5, J = 8.4 Hz). ^13^C NMR (100 MHz, CD_3_OD-*d*_4_): *δ* 27.3 (C-13′), 37.8 (C-1′, 1‴, 4′), 53.1 (C-4‴), 53.8, 54.1 (C-7′, 8′, 10′, 11′), 56.2 (C-12′), 58.0 (C-5′), 68.2 (C-14′), 102.7 (C-8), 104.2 (C-8″), 112.1 (C-3, 3″), 112.3 (C-5a″), 114.0 (C-5a), 114.2 (C-6), 115.5 (C-6″), 127.5 (C-5, 5″), 151.2 (C-4), 152.8 (C-4″), 156.9 (C-8a″), 157.3 (C-8a), 163.2, 164.0 (C-2, 2″), 164.1 (C-7″), 165.5 (C-7), 171.1 (C-2‴), 171.4 (C-2′). Anal. calcd. for C_32_H_35_N_3_O_9_: C, 63.46; H, 5.82; N, 6.94. Found: C, 63.40; H, 5.70; N, 6.90.

Methyl 2-(7-{2-[4-(2-{[2-(7-hydroxy-2-oxo-2*H*-4-chromenyl)acetyl]amino}ethyl)piperazino]buthoxy}-2-oxo-2*H*-4-chromenyl)acetate (**12b**): The crude product purified by column chromatography on silica gel using a mixture of methanol and chlorophorm yielded yellow solid in 66%. Mp. 85–88 °C. ^1^H NMR (400 MHz, CD3OD-*d*_4_, *δ* ppm): *δ* 1.65–1.73 (m, 2H, CH_2_, H-13′), 1.80–1.85 (m, 2H, CH_2_, H-14′), 2.41–2.55 (m, 12H, 6 × CH_2_, H-5′, 7′, 8′, 10′–12′), 3.31 (s, 4H, 2 × CH_2_, H-1′, 1‴), 3.34 (dt, 2H, CH_2_, H-4′, J = 6.4; 12.8 Hz), 3.71 (s, 3H, CH_3_, H-4‴), 4.11 (t, 2H, CH_2_, H-15′, J = 6.0 Hz), 6.15 (s, 2H, 2 × CH, H-3, 3″), 6.63 (d, 1H, CH, H-8″, J = 2.0 Hz), 6.75 (dd, 1H, CH, H-6″, J = 1.6; 8.4 Hz), 6.92 (d, 1H, CH, H-8, J = 2.4 Hz), 6.94 (dd, 1H, CH, H-6, J = 2.4; 9.2 Hz), 7.55 (d, 1H, CH, H-5″, J = 8.8 Hz), 7.60 (s, 1H, CH, H-5, J = 8.4 Hz). ^13^C NMR (100 MHz, CD_3_OD-*d*_4_): *δ* 27.0 (C-13′), 28.2 (C-14′), 37.8 (C-1′, 1‴, 4′), 53.2 (C-4‴), 53.7, 54.0 (C-7′, 8′, 10′, 11′), 57.9, 59.2 (C-5′, 12′), 69.7 (C-15′), 102.6 (C-8), 104.4 (C-8″), 111.7 (C-3, 3″), 112.0 (C-5a″), 113.9 (C-5a), 114.3 (C-6), 115.9 (C-6″), 127.4, 127.5 (C-5, 5″), 151.3 (C-8a), 152.9 (C-8a″), 156.9 (C-4), 157.3 (C-4″), 163.3; 164.1 (C-2, 2″) 164.2, 165.6 (C-7, 7″), 171.1 (C-2′), 171.5 (C-2‴). Anal. calcd. for C_34_H_39_N_3_O_9_: C, 64.44 (64.30); H, 6.20 (6.18); N, 6.63 (6.59). Found: C, 64.30; H, 6.18; N, 6.59.

Methyl 2-[7-({7-[4-(2-{[2-(7-hydroxy-2-oxo-2*H*-4-chromenyl)acetyl]amino}ethyl)piperazino]hexyl}oxy)-2-oxo-2*H*-4-chromenyl]acetate (**12c**): The crude product purified by column chromatography on silica gel using a mixture of methanol and chlorophorm yielded yellow solid in 32%. Mp. 83–86 °C. ^1^H NMR (600 MHz, CD_3_OD-*d*_4_, *δ* ppm): δ 1.37–1.42 (m, 2H, CH_2_, H-14′), 1.51–1.57 (m, 4H, 2 × CH_2_, H-13′, 15′), 1.80–1.85 (m, 2H, CH_2_, H-16′), 2.37 (t, 2H, CH_2_, H-12′, J = 8.4 Hz), 2.44 -2,63 (m, 10H, 5 × CH_2_, H-5′, 7′, 8′, 10′, 11′), 3.33 (s, 4H, 2 × CH_2_, H-1′, 1‴), 3.34–3.35 (m, 2H, CH_2_, H-4′), 3.72 (s, 3H, OCH_3_, H-4‴), 4.09 (t, 2H, CH_2_, H-17′, J = 6.6 Hz), 6.16 (s, 2H, 2 × CH, H-3, 3″), 6.64 (d, 1H, CH, H-8″, J = 2.4 Hz), 6.75 (dd, 1H, CH, H-6″, J = 2.4; 8.4 Hz), 6.91 (d, 1H, CH, H-8, J = 2.4 Hz), 6.94 (dd, 1H, CH, H-6, J = 2.4; 9.0 Hz), 7.56 (d, 1H, CH, H-5″, J = 9.0 Hz), 7.60 (d, 1H, CH, H-5, J = 8.4 Hz). ^13^C NMR (150 MHz, CD_3_OD-*d*_4_): *δ* 27.0 (C-15′), 27.2 (C-13′), 28.3 (C-14′), 30.0 (C-16′), 37.6 (C-4′, 1‴), 53.0 (C-4‴), 53.5 (C-7′, 11′), 53.9 (C-8′,10′), 57.8 (C-5′), 59.5 (C-12′), 69.7 (C-17′), 102.4 (C-8), 104.2 (C-8″), 111.5 (C-3, 3″), 111.9 (C-5a″), 113.7 (C-5a), 114.1 (C-6), 115.8 (C-6″), 127.3 (C-5, 5″), 151.1 (C-4), 152.7 (C-4″), 156.8 (C-8a), 157.3 (C-8a″), 163.2 (C-2), 163.2 (C-2″), 164.1 (C-7), 166.2 (C-7″), 170.1 (C-2′), 171.0 (C-2‴). Anal. calcd. for C_35_H_41_N_3_O_9_: C, 64.90 (64.86); H, 6.38 (6.30); N, 6.49 (6.38). Found: C, 64.86; H, 6.30); N, 6.38.

Methyl 2-[7-({2-[4-(2-{[2-(7-hydroxy-2-oxo-2*H*-4-chromenyl)acetyl]amino}ethyl)piperazino]heptyl}oxy)-2-oxo-2*H*-4-chromenyl]acetate (**12d**): The crude product purified by column chromatography on silica gel using a mixture of methanol and chlorophorm yielded yellow solid in 54%. Mp. 73–76 °C. ^1^H NMR (400 MHz, CD_3_OD-*d*_4_, *δ* ppm): *δ* 1.30–1.45 (m, 6H, 3 × CH_2_, H-14′–16′), 1.48–1.56 (m, 2H, CH_2_, H-13′), 1.79–1.85 (m, 2H, CH_2_, H-17′), 2.34–2.38 (m, 2H, CH_2_, H-12′), 2.40–2.60 (m, 10H, 5 × CH_2_, H- 5′, 7′, 8′, 10′, 11′), 3.31 (s, 4H, 2 × CH_2_, H-1′, 1‴), 3.35 (dt, 2H, CH_2_, H-4′, J = 6.4 Hz), 3.72 (s, 3H, OCH_3_, H-4‴), 4.09 (t, 2H, CH_2_, H-18′, J = 6.8 Hz), 6.19 (s, 2H, 2 × CH, H-3, 3″), 6.66 (d, 1H, CH, H-8″, J = 2.4 Hz), 6.76 (dd, 1H, CH, H-6″, J = 2.4; 8.8 Hz), 6.91 (d, 1H, CH, H-8, J = 2.4 Hz), 6.94 (dd, 1H, CH, H-6, J = 2.4; 8.8 Hz), 7.57 (d, 1H, CH, H-5″, J = 9.2 Hz), 7.61 (d, 1H, CH, H-5, J = 9.2 Hz). ^13^C NMR (100 MHz, CD_3_OD-*d*_4_): *δ* 27.1, 27.4, 28.7, 30.2, 30.3 (C-13′–17′), 37.8 (C-1′, 1‴, 4′), 53.1 (C-4‴), 53.7, 54.0 (C-7′, 8′, 10′, 11′), 58.0 (C-5′), 59.8 (C-12′), 69.9 (C-18′), 102.6 (C-8), 104.2 (C-8″), 112.0 (C-5a″), 112.3 (C-3, 3″), 113.9 (C-5a), 114.2 (C-6), 115.5 (C-6″), 127.5 (C-5, 5″), 151.2 (C-8a, 8a″), 156.9, 157.3 (C-4, 4″), 163.2, 163.9 (C-2, 2″), 164.3 (C-7), 165.7 (C-7″), 171.0, 171.4 (C-2′, 2‴). Anal. calcd. for C_36_H_43_N_3_O_9_: C, 65.34 (65.29); H, 6.55 (6.50); N, 6.35 (6.29). Found: C, 65.29; H, 6.50; N, 6.29.

Methyl 2-[7-({2-[4-(2-{[2-(7-hydroxy-2-oxo-2*H*-4-chromenyl)acetyl]amino}etyl)piperazino]oktyl}oxy)-2-oxo-2*H*-4-chromenyl]acetate (**12e**): The crude product purified by column chromatography on silica gel using a mixture of methanol and chlorophorm yielded yellow solid in 68%. Mp. 70–73 °C. ^1^H NMR (400 MHz, CD_3_OD-*d*_4_, *δ* ppm): *δ* 1.28–1.55 (m, 10H, 5 × CH_2_, H-13′-17′), 1.75–1.85 (m, 2H, CH_2_, H-18′), 2.36 (t, 2H, CH_2_, H-12′, J = 7.8 Hz), 2.40–2.60 (m, 10H, 5 × CH_2_, H-5′, 7′, 8′, 10′, 11′), 3.31 (s, 4H, 2 × CH_2_, H-1′, 1‴), 3.35 (dt, 2H, CH_2_, H-4′, J = 12.4; 6.4 Hz), 3.72 (s, 3H, OCH_3_, H-4‴), 4.07 (t, 2H, CH_2_, H-19′, J = 6.0 Hz), 6.19 (s, 2H, 2 × CH, H-3, 3″), 6.66 (d, 1H, CH, H-8″, J = 2.0 Hz), 6.77 (dd, 1H, CH, H-6″, J = 2.0; 8.4 Hz), 6.90 (d, 1H, CH, H-8, J = 2.0 Hz), 6.92 (dd, 1H, CH, H-6, J = 2.0; 9.0 Hz), 7.58 (d, 1H, CH, H-5″, J = 8.4 Hz), 7.61 (d, 1H, CH, H-5, J = 6.4 Hz). ^13^C NMR (100 MHz, CD3OD-*d*_4_): *δ* 27,1, 27.3, 28.6, 30.2, 30.4, 30.6 (C-13′–18′), 37.8 (C-1′, 1‴, 4′), 53.2 (C-4‴), 53.6, 54.0 (C-7′, 8′, 10′, 11′), 57.9 (C-5′), 59.8 (C-12′), 69.9 (C-19′), 102.6 (C-8), 104.2 (C-8″), 112.2 (C-5a″), 112.4 (C-3, 3″), 113.8 (C-5a), 114.3 (C-6), 115.4 (C-6″), 127.5 (C-5, 5″), 151.3 (C-8a), 152.8 (C-8a″), 156.9, 157.2 (C-4, 4″), 163.3; 163.9 (C-2, 2″), 164.2 (C-7″), 165.2 (C-7), 171.1, 171.5 (C-2′, 2‴). Anal. calcd. for C_37_H_45_N_3_O_9_: C, 65.76 (65.71); H, 6.71 (6.90); N, 6.22 (6.19). Found: C, 65.71; H, 6.90; N, 6.19.

### 3.4. Tests of Anti-AD Action of Biscoumarine Derivatives

#### 3.4.1. In Vitro Anti-Cholinesterase Assay

The AChE/BChE inhibitory activity of the tested drugs was determined using Ellman′s method and is expressed as IC_50_, i.e., the concentration at which cholinesterase activity had been reduced by 50%. Human recombinant AChE (EC 3.1.1.7), 5,5′-dithiobis(2-nitrobenzoic acid) (Ellman′s reagent, DTNB), phosphate buffer (PB, 0.1 M, pH 7.4), and acetylthiocholine (ATC) and butyrylthiocholine (BTC), were purchased from Sigma-Aldrich, Prague, Czech Republic. Human plasma was used as a source of BChE and was prepared from heparinized human blood. The blood was centrifuged for 20 min (4 °C, 2300× *g*) on a Hettich Universal 320R centrifuge. The plasma was separated and stored at 80 °C. Polystyrene Nunc 96-well microplates with a flat bottom shape (ThermoFisher Scientific, Waltham, MA, USA) were used for measuring purposes.

All of the assays were carried out in a 0.1 M phosphate buffer, pH 7.4. The assay medium (100 µL) consisted of 40 µL of 0.1 M phosphate buffer (pH 7.4), 20 µL of 0.01 M DTNB, 10 µL of the enzyme, and 20 µL of 0.01 M substrate ATC/BTC iodide solution).

Inhibitor solutions at a concentration range of 10^−4^–10^−9^ M were prepared, and the IC_50_ values were calculated. The tested compounds were preincubated for 5 min prior to use. The reaction was initiated immediately upon addition of 20 µL of the substrate. The activity was determined by measuring the increase in absorbance at 412 for AChE/BChE at 37 °C at 2 min intervals using a multi-mode microplate reader Synergy 2 (BioTec, Winooski, VT, USA). Each concentration was assayed in triplicate. Software GraphPad Prism 5 (San Diego, CA, USA) was used in the evaluation of the statistical data.

#### 3.4.2. Molecular Modelling Studies

The structure of *h*AChE was gained from RCSB Protein Data Bank—PDB ID: 4EY7 [52]. All receptor structures were prepared by DockPrep function of UCSF Chimera (version 1.4) and converted to pdbqt-files by AutodockTools (v. 1.5.6) [62,63]. Flexible residues selection was based on previous experience with *h*AChE selecting the spherical region around the binding cavity [64,65,66]. Three-dimensional structures of ligands were built by Open Babel (v. 2.3.1), minimized by Avogadro (v 1.1.0), and converted to pdbqt-file format by AutodockTools [67]. The docking calculations were made by Autodock Vina (v. 1.1.2) with the exhaustiveness of 8 [68]. Calculation was repeated 20 times for each ligand and receptor, and the best-scored result was selected for manual inspection. The visualization of enzyme–ligand interactions was prepared using the PyMOL Molecular Graphics System, Version 2.4.1 Schrödinger, LLC, Mannheim, Germany. Two-dimensional figures were generated with Maestro 12.3 (Schrödinger Release, Schrödinger, LLC, New York, NY, USA, 2020).

#### 3.4.3. Determination of In Vitro BBB Permeation

The parallel artificial membrane permeability assay (PAMPA) is used as a non-cell-based in vitro assay to predict BBB penetration and is carried out in a coated 96-well membrane filter. In our assays, the filter membrane of the donor plate was coated with PBL (Polar Brain Lipid, Avanti, Denver, CO, USA) in dodecane (4 µL of 20 mg mL^−1^ PBL in dodecane), and the acceptor well was filled with 300 µL of phosphate buffered saline (PBS) (pH 7.4; VA). The tested compounds were first dissolved in DMSO, and the resulting mixture was diluted with PBS (pH 7.4) to reach the final concentration (40–100 µM) in the donor well. The concentration of DMSO did not exceed 0.5% (*v/v*) in the donor solution. Three hundred microliters of the donor solution (VD) were added to the donor wells, and the donor filter plate was carefully placed onto the acceptor plate to ensure that the coated membrane was “in contact” with both the donor solution and the acceptor buffer. The test compound diffused from the donor well through the polar brain lipid membrane (area = 0.28 cm^2^) to the acceptor well. The concentration of the tested compound in both the donor and acceptor wells was assessed after 3, 4, 5, and 6 h of incubation, respectively, in quadruplicate using a UV plate reader Synergy HT (Biotek, Winooski, VT, USA) at the maximum absorption wavelength of each compound. Additionally, solutions at the theoretical equilibrium of the given compound (i.e., the theoretical concentration if the donor and acceptor compartment were simply combined) were also prepared. The concentrations of the compounds in the donor and acceptor wells and the equilibrium concentration were calculated from the standard curve and are expressed as the permeability (*Pe*) according the equation [51]:*Pe* = C × −ln(1 − [drug]_acceptor_/[drug]_equilibrium_)
where
C = [(V_D_ × V_A_)/((V_D_ + V_A_) × Area × Time)]

### 3.5. In Vitro Antiproliferative Activity and Intracellular Localization

#### 3.5.1. Cell Culture and Treatment

Human lung carcinoma cell line A549 and CCD-18Co colon fibroblasts were purchased from the American Type Culture Collection (ATCC, Rockville, MD, USA). The A549 cells were cultured in a complete RPMI-1640 medium (Sigma-Aldrich, St. Louis, MO, USA), and CCD-18Co cells were cultured in an MEM medium (PAN-Biotech GmbH, Aidenbach, Germany). The media were supplemented with 10% fetal bovine serum (FBS; Biosera, Nuaille, France) and antibiotics (1% Antibiotic-Antimycotic 100× and 50 × 10^−3^ g l^−1^ gentamicin; Biosera) at 37 °C, 95% humidity, and 5% CO_2_.

Prior to the selected treatments, cells were seeded on 12-well *μ*-Chamber slides (ibidi GmbH, Martinsried, Germany) and 6 and/or 96-well plates (TPP, Trasadingen, Switzerland) and left to settle for 24 h. The biscoumarin derivative solutions (at concentrations ranging from 10–100 µM) were then added to cells for 24 or 48 h, and analysis was subsequently performed.

#### 3.5.2. MTT Assay

MTT assays were performed in order to evaluate changes in the metabolic activity of cells that had occurred as a consequence of treatment with the biscoumarin derivatives. A549 cells (15 × 10^3^ cells/cm^2^) and CCD-18Co (15 625 × 10^3^ cells/cm^2^) were seeded in 96-well microplates. The cells were treated for 24 and 48 h with different concentrations (10, 25, 35, 50, 65, 75, and 100 µM) of the derivatives. After the treatment, the MTT (3-(4,5-dimethylthiazol-2-yl)-2,5-diphenyltertrazolium bromide) solution in PBS (5 mg mL^−1^) was added to each well. The reaction was stopped after 4 h incubation, and the formazan was dissolved by the addition of sodium dodecyl sulfate (SDS) at a final concentration of 3.3%. The absorbance of formazan (*λ* = 584 nm) was measured using a BMG FLUOstar Optima spectrometer (BMG Labtech GmbH, Offenburg, Germany). The results were evaluated as percentages of the absorbance of the untreated control. Results are presented as the average percentage of cells from three independent experiments. IC_50_ values for the derivatives were extrapolated from a sigmoidal fit (dose–response curve) to the metabolic activity data using OriginPro 8.5.0 SR1 (OriginLab Corp., Northampton, MA, USA).

#### 3.5.3. Quantification of Cell Number and Viability

For the assessment of total cell numbers and viability within individual experimental groups, floating and adherent cells were harvested after treatment with the studied derivatives and evaluated using a Bürker chamber with eosin staining. A549 cells were plated at a density of 135 × 10^3^ cell/well into a 6-well plate and treated with different concentrations of derivatives (35 and 65 µM) for 24 hrs. The total cell number was expressed as a percentage of the total cell number of the untreated control. Viability was expressed as a percentage of viable, eosin negative cells. The results are presented as the average percentage of cells from three independent experiments.

#### 3.5.4. Colony-Forming Assay

For the colony-forming assay, floating and adherent cells were harvested together 24 h after treatment with the studied derivatives (35 and 65 µM). The cells were then counted using a Bürker chamber with eosin staining, and 500 viable cells per well were seeded in 6-well plates. After 7 days of cultivation under standard conditions, the cells in the plates were fixed and stained with 1% methylene blue dye in methanol. Visualized colonies were scanned and counted, and the results were evaluated as percentages of the untreated control. The results are presented as the average percentage of colonies from three independent experiments.

#### 3.5.5. Cell Cycle Analysis

For flow cytometric analysis of the cell cycle distribution, floating and adherent cells were harvested together 24 h after treatment with the derivatives (35 and 65 µM), washed in cold PBS, fixed in cold 70% ethanol, and stored overnight at −20 °C. Prior to analysis, the cells were washed twice in PBS, resuspended in a staining solution (0.1% Triton X-100, 0.137 g L^−1^ ribonuclease A and 0.02 g L^−1^ propidium iodide—PI), and incubated in dark conditions at RT for 30 min. The DNA content was analyzed using a BD FACSCalibur flow cytometer (Becton Dickinson, San Jose, CA, USA) with a 488 nm argon-ion excitation laser, and fluorescence was detected using a 585/42 nm band-pass filter (FL-2). ModFit 3.0 software (Verity Software House, Topsham, ME, USA) was used to generate DNA content frequency histograms and to quantify the percentage of cells in the individual cell cycle phases. The results are presented as the average ratio of cells in the individual phase to all cells from three independent experiments.

#### 3.5.6. Immunofluorescence Labelling

The studied biscoumarin derivatives were added to cells seeded on 12-well *μ*-Chamber slides for 24 h. After cultivation with the derivatives, the samples (both the control without the derivatives and experimental groups) were washed with a 0.1 M PBS solution buffer to remove all non-cell-absorbed derivative residues and fixed with 4% paraformaldehyde for 20 min at RT. Alternatively, cells were incubated with Mitotracker Red (0.5 µM; Thermo Fisher Scientific, Waltham, MA, USA) for 30 min at 37 °C, washed with 0.1 M PBS, and fixed with 4% paraformaldehyde for 20 min at RT. For the quantification of proliferative activity and programmed cell death, samples were incubated overnight at 4 °C with the mouse anti-Ki67 antibody (NCL-L-Ki67-MM1, 1:100, Leica Biosystems, Wetzlar, Germany) and rabbit anti-cleaved PARP (cat. no. 94885, 1:100, Cell Signaling, Danvers, MA, USA) diluted in 0.1 M PBS containing 1% normal donkey serum (Jackson Immunoresearch, Cambridgeshire, Great Britain) and 0.3% Triton-X 100. After incubation with the primary antibodies, the sections were washed in 0.1 M PBS and incubated with donkey anti-rabbit Alexa Fluor 488 antibody (ab150073, 1:500, Abcam, Cambridge, UK) and donkey anti-mouse Alexa Fluor 555 antibody (ab150109, 1:500, Abcam, Cambridge, UK) diluted in 0.1 M PBS containing 1% normal donkey serum and 0.3% Triton-X 100 for 2 h at RT. Sections were washed in 0.1 M PBS, counterstained with DRAQ5 (1:1000; Thermo Scientific, Waltham, MA, USA) for general nuclear staining for 30 min at RT, and mounted using ProLong^®^ Gold Antifade Reagent (Invitrogen, Carlsbad, CA, USA). All samples were washed with PBS, mounted, and cover slipped with Prolong Gold Antifade media (Thermo Fisher Scientific).

#### 3.5.7. Confocal Microscopy

All samples were evaluated using a Leica TCS SP5X confocal system (Leica Microsystems, Mannheim, Germany). The biscoumarin derivatives were visualized in cells using a 405 nm laser, and the fluorescence was captured in the range of 420–480 nm with identical exposure parameters for all samples. MitoTracker, antibodies, and Draq5 were visualized with White Light laser at appropriate wavelengths. Microphotographs were taken with a 40× objective lens (NA = 0.75; voxel size 223 × 223 nm) and a 63× oil lens (NA = 1.4; voxel size 171 × 171 nm); the images were captured and analyzed using LAS AF software (Leica Microsystems, Wetzlar, Germany).

#### 3.5.8. Image Analyses

Densitometric analysis was performed using Ellipse 2 software (ViDiTo, Košice, Slovak Republic). The images were desaturated prior to analysis. For calibration purposes, the background signal in each sample (i.e., the area without the presence of cells) was set as 0, and the signal from overexposed areas (R = 255; G = 255; B = 255) was set as 100. Densitometric analysis of signal from the cell body area was performed on 20 cells per sample (cells incubated with particular derivatives or control, *n* = 3 samples per each experimental group) and was expressed as median fluorescence intensity (MFI).

The proliferative activity of cells incubated with the studied derivatives was expressed as a Ki67 index. In all samples (*n* = 3 samples per each experimental group), the number of Ki67^+^ nuclei was calculated in groups of 100 randomly selected cells. The ratio of Ki67^+^ nuclei to Draq5^+^ nuclei was multiplied by 100 for improved visualization and was used to obtain the percentage of dividing cells to all cells (Ki67 index).

#### 3.5.9. Statistical Analysis

The obtained data were analyzed using a one-way ANOVA with Tukey′s post-test and are expressed as the mean ± standard deviation (S.D.) of at least three independent experiments. The experimental groups treated with derivatives were compared with the control group: (*): *p* ˂ 0.05, (**): *p* ˂ 0.01, (***): *p* ˂ 0.001.

## 4. Conclusions

A series of new biscoumarin derivatives **12a**–**e** were synthesized and evaluated for their biological activity. Newly synthesized biscoumarins are endowed with a promising profile of biological activities that include inhibition of cholinesterases, reduction in the proliferation of cancer cells without any toxic effect against normal fibroblasts, crossing the BBB, and dual-binding site character in the gorge of *h*AChE. The results suggest that biscoumarin **12d** displayed inhibitory activity against both *h*AChE and *h*BChE (IC_50_ = 6.30 µM, 49 µM) in comparison to compound **12e**. The antiproliferative activity of biscoumarin derivatives **12a**–**e** with increasing length of alkoxy chain increased. The compounds **12d** and **12e** displayed the best in vitro antiproliferative activity on the A549 cancer cell line (IC_50_ = 58 µM, 35 µM) without significant toxicity on normal fibroblast cells CCD-18Co (IC_50_ > 100 µM). The compound **12e** after 24 h incubation suppressed A549 cells colony formation, arrested cell cycle via antiproliferative effect, and accumulation in cytoplasm. Furthermore, compound **12e** showed a high probability to cross the BBB, perhaps, it might be potential compound to alter of cancer cells in brain, too. The results support the potential of **12e** as therapeutic agents against cancer. None showed a significant negative effect of derivative **12d** on metabolism of normal cells, good inhibition activity, and selectivity against *h*AChE, and also, prediction of the ligand’s dual-binding site character in the gorge of *h*AChE by molecular modeling give the possibility of **12d** derivative for application as potential preventing or therapeutic agents against AD. In addition, given structural similarity **12d** to KA- 672, which possesses affinity to 5-HT_1A_, 5-HT_7_, D_2_, and D_3_ receptors, we may speculate that the compounds reported herein, especially **12e** with high probability of CNS availability, might be applicable in other CNS disorders such as schizophrenia.

## Figures and Tables

**Figure 1 ijms-22-03830-f001:**
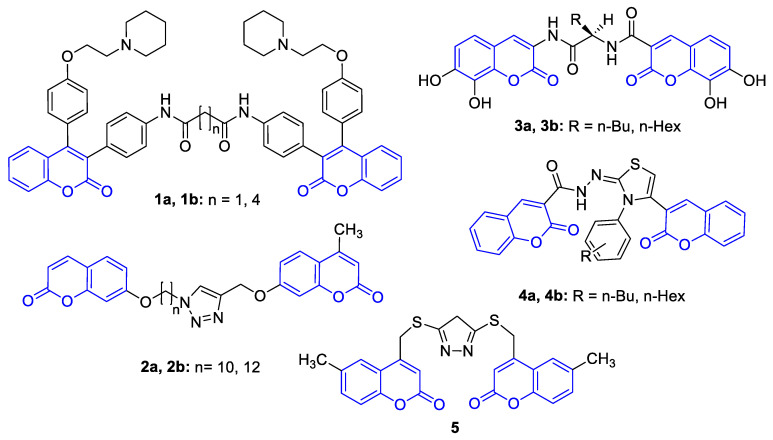
Chemical structure of selected multi-anticancer-directed ligands **1**–**5** bearing a coumarin moiety.

**Figure 2 ijms-22-03830-f002:**
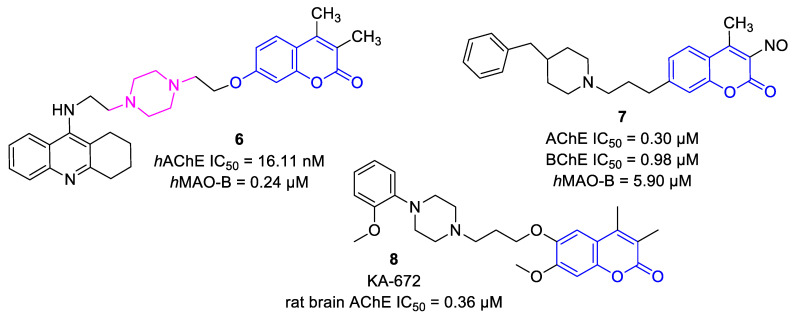
Chemical structure of selected anti-Alzheimer-directed ligands bearing a coumarin moiety **6**–**8**.

**Figure 3 ijms-22-03830-f003:**
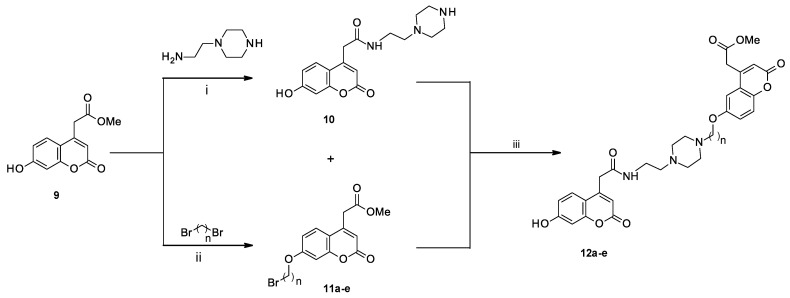
Synthesis of target biscoumarin derivatives **12a**–**e**. Reagents and conditions: (i) CH_3_CN, reflux, 20 h, 94%, (ii) *n* = 3, 4, 6–8, K_2_CO_3_, dry acetone, reflux, 4 h, 38–73%, (iii) K_2_CO_3_, KI, dry acetone, reflux, 27–33 h, 30–68%.

**Figure 4 ijms-22-03830-f004:**
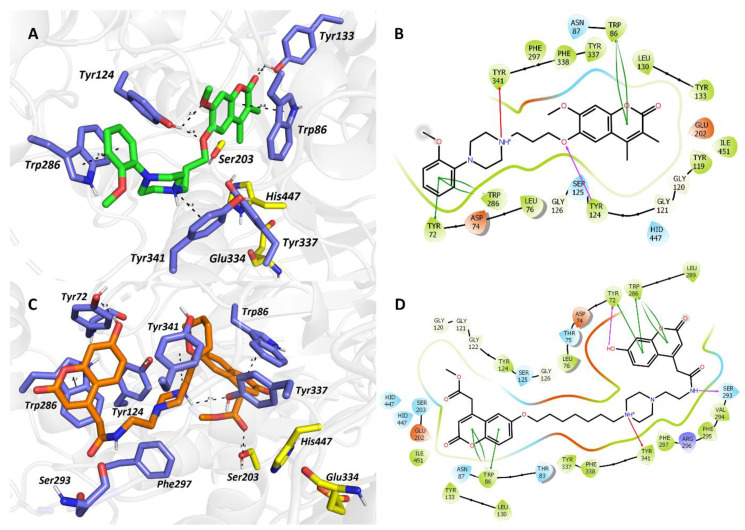
The top-scored docking poses of ligands KA-672 (**A**,**B**) and **12d** (**C**,**D**) in the *h*AChE active site (PDB ID: 4EY7). The ligands are displayed in green (KA-672, (**A**)) and orange (**12d**, (**C**)); important amino acid residues responsible for ligand anchoring are shown in dark blue. Catalytic triad residues are displayed in yellow. Important interactions are rendered by black dashed lines; distances are measured in angstroms (Å). The rest of the receptor is displayed in light-grey cartoon conformation (**A**,**C**). Figures (**A**,**C**) were created with the PyMOL Molecular Graphics System, Version 2.4.1, Schrödinger, LLC. Two-dimensional figures (**B**,**D**) were generated with Maestro 12.3 (Schrödinger Release, Schrödinger, LLC, New York, NY, USA, 2020).

**Figure 5 ijms-22-03830-f005:**
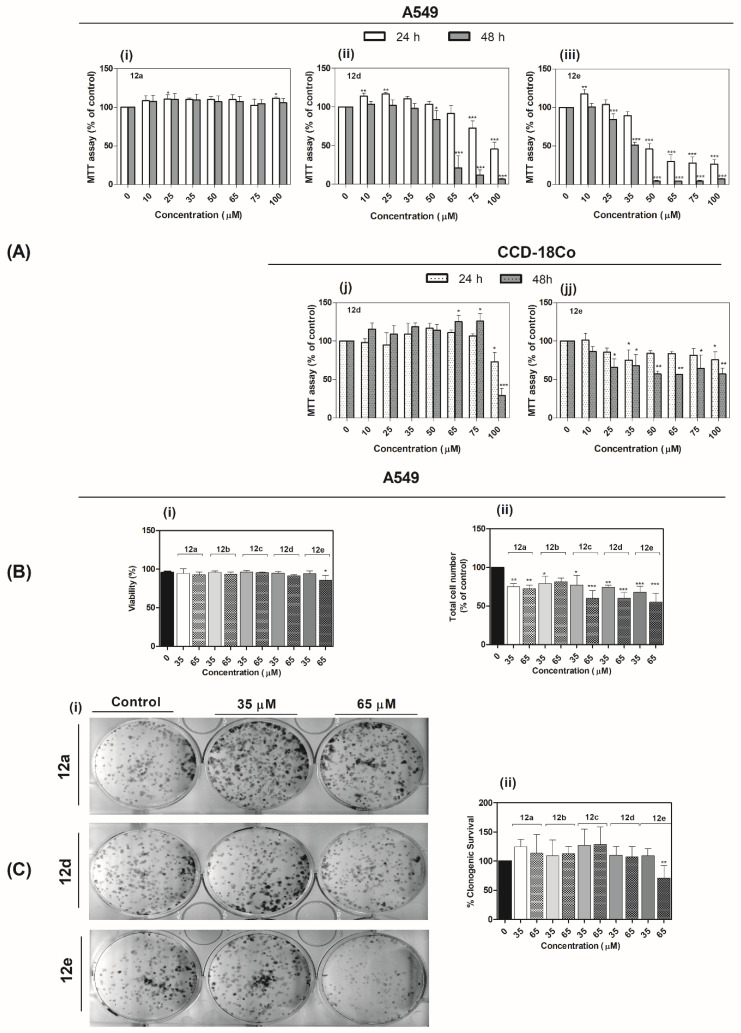
In vitro antiproliferative activity. (**A**): Effects of derivatives **12a** (i), **12d** (ii/j), and **12e** (iii/jj) on A549/CCD-18Co cell line metabolic activity evaluated by MTT assay. The cells were treated with the indicated concentrations of the compounds for 24 and 48 h. Statistical significance * *p* < 0.05; ** *p* < 0.01; *** *p* < 0.001 for each experimental group compared to the untreated cells. (**B**): Effect of derivatives **12a**–**e** on cell viability (i) and total cell number (ii) of A549 cell line. The viability and total cell number were evaluated 24 h after derivatives addition and are expressed as percentage of the viable, eosin negative cells or as a percentage of the total cell number of untreated control, respectively. Statistical significance * *p* < 0.05; ** *p* < 0.01; *** *p* < 0.001 for each experimental group compared to the untreated cells. (**C**): Effect of derivatives **12a**–**e** on A549 cells’ ability to form colonies. (i) Representative image of colonies in both the presence and absence of compounds (**12a**, **12d**, and **12e**) treatment. (ii) Data represent the percentage of colony-forming ability presented as the mean values ± S.D. of three independent experiments. Statistical significance (**): *p* ˂ 0.01 for experimental group compared to the untreated cells.

**Figure 6 ijms-22-03830-f006:**
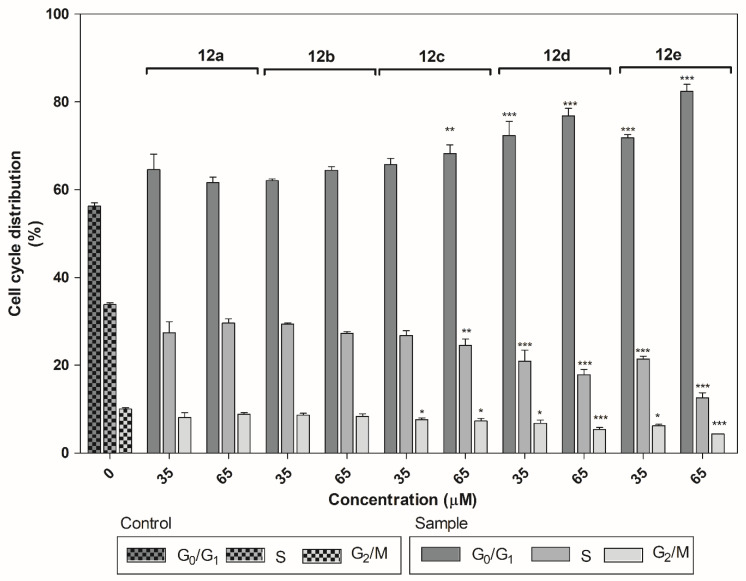
Effect of biscoumarin derivatives **12a**–**e** on cell cycle distribution in A549 cells. The cells were treated with the indicated concentrations of compounds for 24 h. Statistical significance * *p* < 0.05; ** *p* < 0.01; *** *p* < 0.001 for each experimental group compared to the untreated cells.

**Figure 7 ijms-22-03830-f007:**
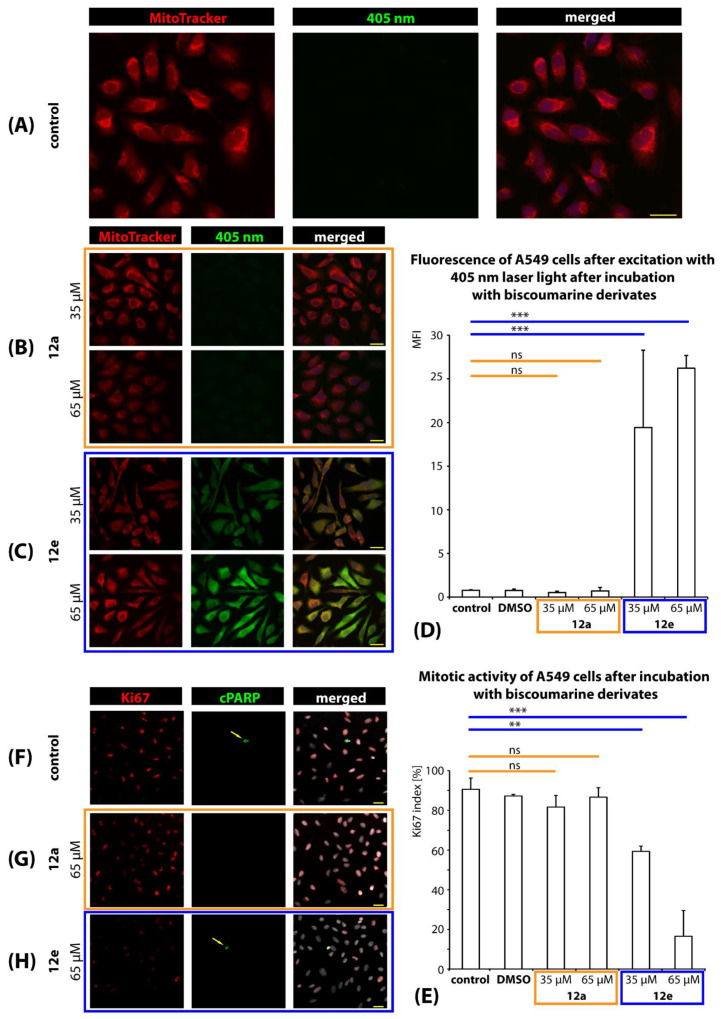
Intracellular localization of derivatives**12a** as representative of **12a**–**d** for comparison with 12e and effect of derivatives **12a** and **12e** on the mitotic activity of A549 cells. Representative microphotographs of A549 cells cultivated in a normal medium without derivatives as control (**A**) or co-cultivated with derivatives **12a** (**B**) and **12e** (**C**), respectively. For the visualization of the cell bodies and nuclei, MitoTracker (red), and Draq5 (blue) stains were used. The studied derivatives would be expected to emit light upon excitation with 405 nm laser light (green). The fluorescence of the A549 cells after co-cultivation with the studied derivatives was quantified with densitometric analysis (**D**). The effect of the derivatives on mitotic activity was quantified as a Ki67 index (**E**) reflecting the percentage of the dividing cells. Representative microphotographs of control (**F**) and cells co-cultivated with derivatives **12a** (**G**) and **12e** (**H**), respectively. Anti-Ki67 antibody labels the nuclei of proliferating cells (red), anti-cleaved PARP is used for the visualization of apoptotic cells (green). Examples of apoptotic cells are highlighted with yellow arrows. Nuclei are stained with Draq5 (gray). Scale bar = 25 µm. Statistical significance for multiple comparisons: ns, nonsignificant; ** *p* < 0.01; *** *p* < 0.001.

**Table 1 ijms-22-03830-t001:** In vitro *h*AChE and *h*BChE inhibitory activity of biscoumarin derivatives **12a**–**e**, reference compounds, selectivity index, and permeability values from the parallel artificial membrane permeability (PAMPA)–BBB assay.

Compound No.	*n*	IC_50_ ± S.E.M. ^a^ (µM) *h*AChE	IC_50_ ± S.E.M. ^a^ (µM) *h*BChE	Selectivity for *h*AChE ^b^	Pe ± S.E.M. (10^−6^ cm s^−1^) ^c^	CNS Predicted Availability ^d^
**12a**	3	>500	>500	-	1.53 ± 0.64	CNS−
**12b**	4	>500	>500	-	0.3 4± 0.10	CNS−
**12c**	6	88.4 ± 12.6	>500	-	2.96 ± 0.73	CNS+/−
**12d**	7	6.30 ± 0.40	49	14.60	2.11 ± 1.19	CNS+/−
**12e**	8	>500	>500	-	13.4 ± 3.56	CNS+
7-MEOTA		15 ± 2.9	21 ± 3.4	1.4	-	-
Tacrine		0.500 ± 0.100	0.023 ± 0.004	0.046	6.0 ± 0.6	CNS+
Donepezil		-	-	-	21.9 ± 2.1	CNS+
Rivastigmine		-	-	-	20.0 ± 2.1	CNS+
Ibuprofen		-	-	-	18.0 ± 4.3	CNS+
Chlorothiazide		-	-	-	1.1 ± 0.5	CNS−
Furosemide		-	-	-	0.2 ± 0.07	CNS−
Ranitidine		-	-	-	0.04 ± 0.02	CNS−
Sulfasalazine		-	-	-	0.09 ± 0.05	CNS−

^a^ The half maximal inhibitory concentration (IC_50_) values are expressed as the mean of at least three experiments (*h*AChE = human recombinant, *h*BChE = human plasma) ± S.E.M. ^b^ Selectivity for *h*AChE is determined as ratio IC_50_ (*h*BChE)/IC_50_ (*h*AChE). ^c^ Permeability coefficient (Pe) values are expressed as the mean of at least three experiments ± S.E.M. ^d^ Classification of the prediction to cross the blood–brain barrier (BBB), CNS+: high BBB permeation predicted with Pe (×10^−6^ cm s^−1^) > 4.0; CNS−: low BBB permeation predicted with Pe (×10^−6^ cm s^−1^) < 2.0; CNS +/−: BBB permeation uncertain with Pe (×10^−6^ cm s^−1^) from 4.0 to 2 [51].

**Table 2 ijms-22-03830-t002:** IC_50_ values of derivatives **12a**–**e** in A549 cancer cell line and CCD-18Co fibroblasts.

Compound No.	*n*	IC_50_ (µM) ^a^				LogP ^b^	LogD ^c^
		A549		CCD-18Co			
		24 h	48 h	24 h	48 h		
**12a**	3	>100	>100	-	-	1.27	0.67
**12b**	4	>100	>100	-	-	1.79	1.01
**12c**	6	>100	>100	-	-	2.68	1.79
**12d**	7	94	58	>100	>100	3.12	2.24
**12e**	8	49	35	>100	>100	3.57	2.68
DMSO (1%)		-	-	96%	98%	-	-

^a^ IC_50_ values are the mean of the half maximal inhibitory concentration. ^b^ Calculate logP—partial coefficient—(lipophilicity) from structure (Percepta Software ACD/Labs). ^c^ Calculate logD—distribution coefficient—(lipophilicity bases on pH for ionizable compounds) from structure (Percepta Software ACD/Labs). - not defined. DMSO, dimethyl sulfoxide.

## Data Availability

The data presented in this study are available on request from the corresponding author.

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
