# Peer review of "Synthesis of New Biscoumarin Derivatives, In Vitro Cholinesterase Inhibition, Molecular Modelling and Antiproliferative Effect in A549 Human Lung Carcinoma Cells"

_ijms, 2021, doi:10.3390/ijms22083830_

Round 1

Reviewer 1 Report

The authors described that Synthesis of new biscoumarin derivatives, in vitro cholinester-2 ase inhibition, molecular modelling and antiproliferative effect 3 in A549 human lung carcinoma cells.
This manuscript involves in in vitro assay, several cell assay and computational modeling studies.
I think, even though all studies were conducted appropriately, the enzyme assay and cell assay showed unreasonable results.
In detail, 12e shows no inhibitory activity against both hAChE and hBChE.
In rathional reasons, the authors should explain 12d along with the potency of hAChE and hBChE.
However, the authors implemented the assay Intracellular localization and mitotic activity of A549 cells with 12e, not 12d.
In contrast, the modeling studies was carried out using 12d.
If the authors would like to claim the good potency of 12e on cell studies according to good permeability, the authors should confirm more high concentration result on enzyme assay, first.
After that, the differency of enzyme and cell results should be discussed.

My request is to see higher concentration results of compound 12e on enzyme assay, at least.
I cannot decide the possibility to publicate this manusctipt without seeing correlation with the result of enzyme assay and cell assay.

Author Response

Poin 1: The authors described that Synthesis of new biscoumarin derivatives, in vitro cholinester-2 ase inhibition, molecular modelling and antiproliferative effect 3 in A549 human lung carcinoma cells.
This manuscript involves in in vitro assay, several cell assay and computational modeling studies.
I think, even though all studies were conducted appropriately, the enzyme assay and cell assay showed unreasonable results.

In detail, 12e shows no inhibitory activity against both hAChE and hBChE.
In rathional reasons, the authors should explain 12d along with the potency of hAChE and hBChE.
However, the authors implemented the assay Intracellular localization and mitotic activity of A549 cells with 12e, not 12d.

Author response 1: We thank the reviewer for this important point. Cytostatic/antiproliferative effects of all studied compounds 12a-12e (including 12d) were analyzed and quantified in sections 2.3.2 and 2.3.3. It was concluded that the compounds 12d and 12e showed the most potent effect on mitotic activity/cell cycle arrest. However, only the compound 12e showed decreased cell viability, indicating cytotoxic effect (Fig. 5B), worth of further analyses (mitotic activity and induction of apoptosis). In another experiments, we also performed the analysis of intracellular localization of all studied compounds 12a-12e (including 12d) on A549 cells using confocal microscopy. However, only the compound 12e showed increased fluorescence of cells indicating the accumulation of the fluorescent chemical derivative, which could be visualized and localized in cells. Therefore, in subsequent analyses of intracellular localization, mitotic activity and induction of apoptosis, we decided to focus only on the compound 12e. This is stated in manuscript (lines 538 - 563). The compound 12a was used only for comparison, which could lead to some misunderstanding. However, if it is necessary, we are ready to perform all analyses also on compound 12d, but in this case we will need significantly more time for revision (approx. 14 days). According to our opinion, the effect of the added analyses on the improvement of manuscript could be questionable, since all of the analyses (except the mitotic activity) should provide negative results.

Poin 2: In contrast, the modeling studies was carried out using 12d.

If the authors would like to claim the good potency of 12e on cell studies according to good permeability, the authors should confirm more high concentration result on enzyme assay, first.
After that, the differency of enzyme and cell results should be discussed.

 Author response 2: It is clear that derivative 12d is a potent inhibitor of hAChE, while 12e not. Therefore it is reasonable to perform the docking studies only for derivative 12d. The IC50 for derivative 12e of  > 500 µM means that due to solubility reason we could not test it in higher concentration. In fact, no effect at such high concentration (500 µM) means that there is definitely no clinical relevance by the AChE inhibition caused by this compound and therefore no rationale for targeting the Alzheimer´s disease. On the other hand, good BBB permeability may be important in targeting other CNS disorders like schizophrenia; and intracellular localization and mitotic activity is suitable for targeting cancer.

We have modified the Abstract and Conclusion accordingly

Poin 3: My request is to see higher concentration results of compound 12e on enzyme assay, at least.
I cannot decide the possibility to publicate this manusctipt without seeing correlation with the result of enzyme assay and cell assay.

Author response 3: We would like to apologize for the misunderstanding. Apparently there was some misunderstanding here, because it was not our intention to prove that the corresponding bis-coumarin derivatives would act as cholinesterase inhibitors and at the same time as anti-cancer drugs. Whereas patients with certain central nervous system (CNS) disorders are known to develop less cancer than expected, than this result that derivative 12e does not inhibit cholinesterases but on the other hand shows good antiproliferative activity is pharmacologically good because there is no reason for a potential cancer drug to inhibit AChE, as this is not necessary for this activity. Basic science increasingly indicates shared molecular mechanisms between cancer and AD and gives weight to key relevant biological theories.

Poin 1: The authors described that Synthesis of new biscoumarin derivatives, in vitro cholinester-2 ase inhibition, molecular modelling and antiproliferative effect 3 in A549 human lung carcinoma cells.
This manuscript involves in in vitro assay, several cell assay and computational modeling studies.
I think, even though all studies were conducted appropriately, the enzyme assay and cell assay showed unreasonable results.

In detail, 12e shows no inhibitory activity against both hAChE and hBChE.
In rathional reasons, the authors should explain 12d along with the potency of hAChE and hBChE.
However, the authors implemented the assay Intracellular localization and mitotic activity of A549 cells with 12e, not 12d.

Author response 1: We thank the reviewer for this important point. Cytostatic/antiproliferative effects of all studied compounds 12a-12e (including 12d) were analyzed and quantified in sections 2.3.2 and 2.3.3. It was concluded that the compounds 12d and 12e showed the most potent effect on mitotic activity/cell cycle arrest. However, only the compound 12e showed decreased cell viability, indicating cytotoxic effect (Fig. 5B), worth of further analyses (mitotic activity and induction of apoptosis). In another experiments, we also performed the analysis of intracellular localization of all studied compounds 12a-12e (including 12d) on A549 cells using confocal microscopy. However, only the compound 12e showed increased fluorescence of cells indicating the accumulation of the fluorescent chemical derivative, which could be visualized and localized in cells. Therefore, in subsequent analyses of intracellular localization, mitotic activity and induction of apoptosis, we decided to focus only on the compound 12e. This is stated in manuscript (lines 538 - 563). The compound 12a was used only for comparison, which could lead to some misunderstanding. However, if it is necessary, we are ready to perform all analyses also on compound 12d, but in this case we will need significantly more time for revision (approx. 14 days). According to our opinion, the effect of the added analyses on the improvement of manuscript could be questionable, since all of the analyses (except the mitotic activity) should provide negative results.

Poin 2: In contrast, the modeling studies was carried out using 12d.

If the authors would like to claim the good potency of 12e on cell studies according to good permeability, the authors should confirm more high concentration result on enzyme assay, first.
After that, the differency of enzyme and cell results should be discussed.

 Author response 2: It is clear that derivative 12d is a potent inhibitor of hAChE, while 12e not. Therefore it is reasonable to perform the docking studies only for derivative 12d. The IC50 for derivative 12e of  > 500 µM means that due to solubility reason we could not test it in higher concentration. In fact, no effect at such high concentration (500 µM) means that there is definitely no clinical relevance by the AChE inhibition caused by this compound and therefore no rationale for targeting the Alzheimer´s disease. On the other hand, good BBB permeability may be important in targeting other CNS disorders like schizophrenia; and intracellular localization and mitotic activity is suitable for targeting cancer.

We have modified the Abstract and Conclusion accordingly

Poin 3: My request is to see higher concentration results of compound 12e on enzyme assay, at least.
I cannot decide the possibility to publicate this manusctipt without seeing correlation with the result of enzyme assay and cell assay.

Author response 3: We would like to apologize for the misunderstanding. Apparently there was some misunderstanding here, because it was not our intention to prove that the corresponding bis-coumarin derivatives would act as cholinesterase inhibitors and at the same time as anti-cancer drugs. Whereas patients with certain central nervous system (CNS) disorders are known to develop less cancer than expected, than this result that derivative 12e does not inhibit cholinesterases but on the other hand shows good antiproliferative activity is pharmacologically good because there is no reason for a potential cancer drug to inhibit AChE, as this is not necessary for this activity. Basic science increasingly indicates shared molecular mechanisms between cancer and AD and gives weight to key relevant biological theories.

Poin 1: The authors described that Synthesis of new biscoumarin derivatives, in vitro cholinester-2 ase inhibition, molecular modelling and antiproliferative effect 3 in A549 human lung carcinoma cells.
This manuscript involves in in vitro assay, several cell assay and computational modeling studies.
I think, even though all studies were conducted appropriately, the enzyme assay and cell assay showed unreasonable results.

In detail, 12e shows no inhibitory activity against both hAChE and hBChE.
In rathional reasons, the authors should explain 12d along with the potency of hAChE and hBChE.
However, the authors implemented the assay Intracellular localization and mitotic activity of A549 cells with 12e, not 12d.

Author response 1: We thank the reviewer for this important point. Cytostatic/antiproliferative effects of all studied compounds 12a-12e (including 12d) were analyzed and quantified in sections 2.3.2 and 2.3.3. It was concluded that the compounds 12d and 12e showed the most potent effect on mitotic activity/cell cycle arrest. However, only the compound 12e showed decreased cell viability, indicating cytotoxic effect (Fig. 5B), worth of further analyses (mitotic activity and induction of apoptosis). In another experiments, we also performed the analysis of intracellular localization of all studied compounds 12a-12e (including 12d) on A549 cells using confocal microscopy. However, only the compound 12e showed increased fluorescence of cells indicating the accumulation of the fluorescent chemical derivative, which could be visualized and localized in cells. Therefore, in subsequent analyses of intracellular localization, mitotic activity and induction of apoptosis, we decided to focus only on the compound 12e. This is stated in manuscript (lines 538 - 563). The compound 12a was used only for comparison, which could lead to some misunderstanding. However, if it is necessary, we are ready to perform all analyses also on compound 12d, but in this case we will need significantly more time for revision (approx. 14 days). According to our opinion, the effect of the added analyses on the improvement of manuscript could be questionable, since all of the analyses (except the mitotic activity) should provide negative results.

Poin 2: In contrast, the modeling studies was carried out using 12d.

If the authors would like to claim the good potency of 12e on cell studies according to good permeability, the authors should confirm more high concentration result on enzyme assay, first.
After that, the differency of enzyme and cell results should be discussed.

 Author response 2: It is clear that derivative 12d is a potent inhibitor of hAChE, while 12e not. Therefore it is reasonable to perform the docking studies only for derivative 12d. The IC50 for derivative 12e of  > 500 µM means that due to solubility reason we could not test it in higher concentration. In fact, no effect at such high concentration (500 µM) means that there is definitely no clinical relevance by the AChE inhibition caused by this compound and therefore no rationale for targeting the Alzheimer´s disease. On the other hand, good BBB permeability may be important in targeting other CNS disorders like schizophrenia; and intracellular localization and mitotic activity is suitable for targeting cancer.

We have modified the Abstract and Conclusion accordingly

Poin 3: My request is to see higher concentration results of compound 12e on enzyme assay, at least.
I cannot decide the possibility to publicate this manusctipt without seeing correlation with the result of enzyme assay and cell assay.

Author response 3: We would like to apologize for the misunderstanding. Apparently there was some misunderstanding here, because it was not our intention to prove that the corresponding bis-coumarin derivatives would act as cholinesterase inhibitors and at the same time as anti-cancer drugs. Whereas patients with certain central nervous system (CNS) disorders are known to develop less cancer than expected, than this result that derivative 12e does not inhibit cholinesterases but on the other hand shows good antiproliferative activity is pharmacologically good because there is no reason for a potential cancer drug to inhibit AChE, as this is not necessary for this activity. Basic science increasingly indicates shared molecular mechanisms between cancer and AD and gives weight to key relevant biological theories.

Poin 1: The authors described that Synthesis of new biscoumarin derivatives, in vitro cholinester-2 ase inhibition, molecular modelling and antiproliferative effect 3 in A549 human lung carcinoma cells.
This manuscript involves in in vitro assay, several cell assay and computational modeling studies.
I think, even though all studies were conducted appropriately, the enzyme assay and cell assay showed unreasonable results.

In detail, 12e shows no inhibitory activity against both hAChE and hBChE.
In rathional reasons, the authors should explain 12d along with the potency of hAChE and hBChE.
However, the authors implemented the assay Intracellular localization and mitotic activity of A549 cells with 12e, not 12d.

Author response 1: We thank the reviewer for this important point. Cytostatic/antiproliferative effects of all studied compounds 12a-12e (including 12d) were analyzed and quantified in sections 2.3.2 and 2.3.3. It was concluded that the compounds 12d and 12e showed the most potent effect on mitotic activity/cell cycle arrest. However, only the compound 12e showed decreased cell viability, indicating cytotoxic effect (Fig. 5B), worth of further analyses (mitotic activity and induction of apoptosis). In another experiments, we also performed the analysis of intracellular localization of all studied compounds 12a-12e (including 12d) on A549 cells using confocal microscopy. However, only the compound 12e showed increased fluorescence of cells indicating the accumulation of the fluorescent chemical derivative, which could be visualized and localized in cells. Therefore, in subsequent analyses of intracellular localization, mitotic activity and induction of apoptosis, we decided to focus only on the compound 12e. This is stated in manuscript (lines 538 - 563). The compound 12a was used only for comparison, which could lead to some misunderstanding. However, if it is necessary, we are ready to perform all analyses also on compound 12d, but in this case we will need significantly more time for revision (approx. 14 days). According to our opinion, the effect of the added analyses on the improvement of manuscript could be questionable, since all of the analyses (except the mitotic activity) should provide negative results.

Poin 2: In contrast, the modeling studies was carried out using 12d.

If the authors would like to claim the good potency of 12e on cell studies according to good permeability, the authors should confirm more high concentration result on enzyme assay, first.
After that, the differency of enzyme and cell results should be discussed.

 Author response 2: It is clear that derivative 12d is a potent inhibitor of hAChE, while 12e not. Therefore it is reasonable to perform the docking studies only for derivative 12d. The IC50 for derivative 12e of  > 500 µM means that due to solubility reason we could not test it in higher concentration. In fact, no effect at such high concentration (500 µM) means that there is definitely no clinical relevance by the AChE inhibition caused by this compound and therefore no rationale for targeting the Alzheimer´s disease. On the other hand, good BBB permeability may be important in targeting other CNS disorders like schizophrenia; and intracellular localization and mitotic activity is suitable for targeting cancer.

We have modified the Abstract and Conclusion accordingly

Poin 3: My request is to see higher concentration results of compound 12e on enzyme assay, at least.
I cannot decide the possibility to publicate this manusctipt without seeing correlation with the result of enzyme assay and cell assay.

Author response 3: We would like to apologize for the misunderstanding. Apparently there was some misunderstanding here, because it was not our intention to prove that the corresponding bis-coumarin derivatives would act as cholinesterase inhibitors and at the same time as anti-cancer drugs. Whereas patients with certain central nervous system (CNS) disorders are known to develop less cancer than expected, than this result that derivative 12e does not inhibit cholinesterases but on the other hand shows good antiproliferative activity is pharmacologically good because there is no reason for a potential cancer drug to inhibit AChE, as this is not necessary for this activity. Basic science increasingly indicates shared molecular mechanisms between cancer and AD and gives weight to key relevant biological theories.

Poin 1: The authors described that Synthesis of new biscoumarin derivatives, in vitro cholinester-2 ase inhibition, molecular modelling and antiproliferative effect 3 in A549 human lung carcinoma cells.
This manuscript involves in in vitro assay, several cell assay and computational modeling studies.
I think, even though all studies were conducted appropriately, the enzyme assay and cell assay showed unreasonable results.

In detail, 12e shows no inhibitory activity against both hAChE and hBChE.
In rathional reasons, the authors should explain 12d along with the potency of hAChE and hBChE.
However, the authors implemented the assay Intracellular localization and mitotic activity of A549 cells with 12e, not 12d.

Author response 1: We thank the reviewer for this important point. Cytostatic/antiproliferative effects of all studied compounds 12a-12e (including 12d) were analyzed and quantified in sections 2.3.2 and 2.3.3. It was concluded that the compounds 12d and 12e showed the most potent effect on mitotic activity/cell cycle arrest. However, only the compound 12e showed decreased cell viability, indicating cytotoxic effect (Fig. 5B), worth of further analyses (mitotic activity and induction of apoptosis). In another experiments, we also performed the analysis of intracellular localization of all studied compounds 12a-12e (including 12d) on A549 cells using confocal microscopy. However, only the compound 12e showed increased fluorescence of cells indicating the accumulation of the fluorescent chemical derivative, which could be visualized and localized in cells. Therefore, in subsequent analyses of intracellular localization, mitotic activity and induction of apoptosis, we decided to focus only on the compound 12e. This is stated in manuscript (lines 538 - 563). The compound 12a was used only for comparison, which could lead to some misunderstanding. However, if it is necessary, we are ready to perform all analyses also on compound 12d, but in this case we will need significantly more time for revision (approx. 14 days). According to our opinion, the effect of the added analyses on the improvement of manuscript could be questionable, since all of the analyses (except the mitotic activity) should provide negative results.

Poin 2: In contrast, the modeling studies was carried out using 12d.

If the authors would like to claim the good potency of 12e on cell studies according to good permeability, the authors should confirm more high concentration result on enzyme assay, first.
After that, the differency of enzyme and cell results should be discussed.

 Author response 2: It is clear that derivative 12d is a potent inhibitor of hAChE, while 12e not. Therefore it is reasonable to perform the docking studies only for derivative 12d. The IC50 for derivative 12e of  > 500 µM means that due to solubility reason we could not test it in higher concentration. In fact, no effect at such high concentration (500 µM) means that there is definitely no clinical relevance by the AChE inhibition caused by this compound and therefore no rationale for targeting the Alzheimer´s disease. On the other hand, good BBB permeability may be important in targeting other CNS disorders like schizophrenia; and intracellular localization and mitotic activity is suitable for targeting cancer.

We have modified the Abstract and Conclusion accordingly

Poin 3: My request is to see higher concentration results of compound 12e on enzyme assay, at least.
I cannot decide the possibility to publicate this manusctipt without seeing correlation with the result of enzyme assay and cell assay.

Author response 3: We would like to apologize for the misunderstanding. Apparently there was some misunderstanding here, because it was not our intention to prove that the corresponding bis-coumarin derivatives would act as cholinesterase inhibitors and at the same time as anti-cancer drugs. Whereas patients with certain central nervous system (CNS) disorders are known to develop less cancer than expected, than this result that derivative 12e does not inhibit cholinesterases but on the other hand shows good antiproliferative activity is pharmacologically good because there is no reason for a potential cancer drug to inhibit AChE, as this is not necessary for this activity. Basic science increasingly indicates shared molecular mechanisms between cancer and AD and gives weight to key relevant biological theories.

Poin 1: The authors described that Synthesis of new biscoumarin derivatives, in vitro cholinester-2 ase inhibition, molecular modelling and antiproliferative effect 3 in A549 human lung carcinoma cells.
This manuscript involves in in vitro assay, several cell assay and computational modeling studies.
I think, even though all studies were conducted appropriately, the enzyme assay and cell assay showed unreasonable results.

In detail, 12e shows no inhibitory activity against both hAChE and hBChE.
In rathional reasons, the authors should explain 12d along with the potency of hAChE and hBChE.
However, the authors implemented the assay Intracellular localization and mitotic activity of A549 cells with 12e, not 12d.

Author response 1: We thank the reviewer for this important point. Cytostatic/antiproliferative effects of all studied compounds 12a-12e (including 12d) were analyzed and quantified in sections 2.3.2 and 2.3.3. It was concluded that the compounds 12d and 12e showed the most potent effect on mitotic activity/cell cycle arrest. However, only the compound 12e showed decreased cell viability, indicating cytotoxic effect (Fig. 5B), worth of further analyses (mitotic activity and induction of apoptosis). In another experiments, we also performed the analysis of intracellular localization of all studied compounds 12a-12e (including 12d) on A549 cells using confocal microscopy. However, only the compound 12e showed increased fluorescence of cells indicating the accumulation of the fluorescent chemical derivative, which could be visualized and localized in cells. Therefore, in subsequent analyses of intracellular localization, mitotic activity and induction of apoptosis, we decided to focus only on the compound 12e. This is stated in manuscript (lines 538 - 563). The compound 12a was used only for comparison, which could lead to some misunderstanding. However, if it is necessary, we are ready to perform all analyses also on compound 12d, but in this case we will need significantly more time for revision (approx. 14 days). According to our opinion, the effect of the added analyses on the improvement of manuscript could be questionable, since all of the analyses (except the mitotic activity) should provide negative results.

Poin 2: In contrast, the modeling studies was carried out using 12d.

If the authors would like to claim the good potency of 12e on cell studies according to good permeability, the authors should confirm more high concentration result on enzyme assay, first.
After that, the differency of enzyme and cell results should be discussed.

 Author response 2: It is clear that derivative 12d is a potent inhibitor of hAChE, while 12e not. Therefore it is reasonable to perform the docking studies only for derivative 12d. The IC50 for derivative 12e of  > 500 µM means that due to solubility reason we could not test it in higher concentration. In fact, no effect at such high concentration (500 µM) means that there is definitely no clinical relevance by the AChE inhibition caused by this compound and therefore no rationale for targeting the Alzheimer´s disease. On the other hand, good BBB permeability may be important in targeting other CNS disorders like schizophrenia; and intracellular localization and mitotic activity is suitable for targeting cancer.

We have modified the Abstract and Conclusion accordingly

Poin 3: My request is to see higher concentration results of compound 12e on enzyme assay, at least.
I cannot decide the possibility to publicate this manusctipt without seeing correlation with the result of enzyme assay and cell assay.

Author response 3: We would like to apologize for the misunderstanding. Apparently there was some misunderstanding here, because it was not our intention to prove that the corresponding bis-coumarin derivatives would act as cholinesterase inhibitors and at the same time as anti-cancer drugs. Whereas patients with certain central nervous system (CNS) disorders are known to develop less cancer than expected, than this result that derivative 12e does not inhibit cholinesterases but on the other hand shows good antiproliferative activity is pharmacologically good because there is no reason for a potential cancer drug to inhibit AChE, as this is not necessary for this activity. Basic science increasingly indicates shared molecular mechanisms between cancer and AD and gives weight to key relevant biological theories.

Poin 1: The authors described that Synthesis of new biscoumarin derivatives, in vitro cholinester-2 ase inhibition, molecular modelling and antiproliferative effect 3 in A549 human lung carcinoma cells.
This manuscript involves in in vitro assay, several cell assay and computational modeling studies.
I think, even though all studies were conducted appropriately, the enzyme assay and cell assay showed unreasonable results.

In detail, 12e shows no inhibitory activity against both hAChE and hBChE.
In rathional reasons, the authors should explain 12d along with the potency of hAChE and hBChE.
However, the authors implemented the assay Intracellular localization and mitotic activity of A549 cells with 12e, not 12d.

Author response 1: We thank the reviewer for this important point. Cytostatic/antiproliferative effects of all studied compounds 12a-12e (including 12d) were analyzed and quantified in sections 2.3.2 and 2.3.3. It was concluded that the compounds 12d and 12e showed the most potent effect on mitotic activity/cell cycle arrest. However, only the compound 12e showed decreased cell viability, indicating cytotoxic effect (Fig. 5B), worth of further analyses (mitotic activity and induction of apoptosis). In another experiments, we also performed the analysis of intracellular localization of all studied compounds 12a-12e (including 12d) on A549 cells using confocal microscopy. However, only the compound 12e showed increased fluorescence of cells indicating the accumulation of the fluorescent chemical derivative, which could be visualized and localized in cells. Therefore, in subsequent analyses of intracellular localization, mitotic activity and induction of apoptosis, we decided to focus only on the compound 12e. This is stated in manuscript (lines 538 - 563). The compound 12a was used only for comparison, which could lead to some misunderstanding. However, if it is necessary, we are ready to perform all analyses also on compound 12d, but in this case we will need significantly more time for revision (approx. 14 days). According to our opinion, the effect of the added analyses on the improvement of manuscript could be questionable, since all of the analyses (except the mitotic activity) should provide negative results.

Poin 2: In contrast, the modeling studies was carried out using 12d.

If the authors would like to claim the good potency of 12e on cell studies according to good permeability, the authors should confirm more high concentration result on enzyme assay, first.
After that, the differency of enzyme and cell results should be discussed.

 Author response 2: It is clear that derivative 12d is a potent inhibitor of hAChE, while 12e not. Therefore it is reasonable to perform the docking studies only for derivative 12d. The IC50 for derivative 12e of  > 500 µM means that due to solubility reason we could not test it in higher concentration. In fact, no effect at such high concentration (500 µM) means that there is definitely no clinical relevance by the AChE inhibition caused by this compound and therefore no rationale for targeting the Alzheimer´s disease. On the other hand, good BBB permeability may be important in targeting other CNS disorders like schizophrenia; and intracellular localization and mitotic activity is suitable for targeting cancer.

We have modified the Abstract and Conclusion accordingly

Poin 3: My request is to see higher concentration results of compound 12e on enzyme assay, at least.
I cannot decide the possibility to publicate this manusctipt without seeing correlation with the result of enzyme assay and cell assay.

Author response 3: We would like to apologize for the misunderstanding. Apparently there was some misunderstanding here, because it was not our intention to prove that the corresponding bis-coumarin derivatives would act as cholinesterase inhibitors and at the same time as anti-cancer drugs. Whereas patients with certain central nervous system (CNS) disorders are known to develop less cancer than expected, than this result that derivative 12e does not inhibit cholinesterases but on the other hand shows good antiproliferative activity is pharmacologically good because there is no reason for a potential cancer drug to inhibit AChE, as this is not necessary for this activity. Basic science increasingly indicates shared molecular mechanisms between cancer and AD and gives weight to key relevant biological theories.

Poin 1: The authors described that Synthesis of new biscoumarin derivatives, in vitro cholinester-2 ase inhibition, molecular modelling and antiproliferative effect 3 in A549 human lung carcinoma cells.
This manuscript involves in in vitro assay, several cell assay and computational modeling studies.
I think, even though all studies were conducted appropriately, the enzyme assay and cell assay showed unreasonable results.

In detail, 12e shows no inhibitory activity against both hAChE and hBChE.
In rathional reasons, the authors should explain 12d along with the potency of hAChE and hBChE.
However, the authors implemented the assay Intracellular localization and mitotic activity of A549 cells with 12e, not 12d.

Author response 1: We thank the reviewer for this important point. Cytostatic/antiproliferative effects of all studied compounds 12a-12e (including 12d) were analyzed and quantified in sections 2.3.2 and 2.3.3. It was concluded that the compounds 12d and 12e showed the most potent effect on mitotic activity/cell cycle arrest. However, only the compound 12e showed decreased cell viability, indicating cytotoxic effect (Fig. 5B), worth of further analyses (mitotic activity and induction of apoptosis). In another experiments, we also performed the analysis of intracellular localization of all studied compounds 12a-12e (including 12d) on A549 cells using confocal microscopy. However, only the compound 12e showed increased fluorescence of cells indicating the accumulation of the fluorescent chemical derivative, which could be visualized and localized in cells. Therefore, in subsequent analyses of intracellular localization, mitotic activity and induction of apoptosis, we decided to focus only on the compound 12e. This is stated in manuscript (lines 538 - 563). The compound 12a was used only for comparison, which could lead to some misunderstanding. However, if it is necessary, we are ready to perform all analyses also on compound 12d, but in this case we will need significantly more time for revision (approx. 14 days). According to our opinion, the effect of the added analyses on the improvement of manuscript could be questionable, since all of the analyses (except the mitotic activity) should provide negative results.

Poin 2: In contrast, the modeling studies was carried out using 12d.

If the authors would like to claim the good potency of 12e on cell studies according to good permeability, the authors should confirm more high concentration result on enzyme assay, first.
After that, the differency of enzyme and cell results should be discussed.

 Author response 2: It is clear that derivative 12d is a potent inhibitor of hAChE, while 12e not. Therefore it is reasonable to perform the docking studies only for derivative 12d. The IC50 for derivative 12e of  > 500 µM means that due to solubility reason we could not test it in higher concentration. In fact, no effect at such high concentration (500 µM) means that there is definitely no clinical relevance by the AChE inhibition caused by this compound and therefore no rationale for targeting the Alzheimer´s disease. On the other hand, good BBB permeability may be important in targeting other CNS disorders like schizophrenia; and intracellular localization and mitotic activity is suitable for targeting cancer.

We have modified the Abstract and Conclusion accordingly

Poin 3: My request is to see higher concentration results of compound 12e on enzyme assay, at least.
I cannot decide the possibility to publicate this manusctipt without seeing correlation with the result of enzyme assay and cell assay.

Author response 3: We would like to apologize for the misunderstanding. Apparently there was some misunderstanding here, because it was not our intention to prove that the corresponding bis-coumarin derivatives would act as cholinesterase inhibitors and at the same time as anti-cancer drugs. Whereas patients with certain central nervous system (CNS) disorders are known to develop less cancer than expected, than this result that derivative 12e does not inhibit cholinesterases but on the other hand shows good antiproliferative activity is pharmacologically good because there is no reason for a potential cancer drug to inhibit AChE, as this is not necessary for this activity. Basic science increasingly indicates shared molecular mechanisms between cancer and AD and gives weight to key relevant biological theories.

Poin 1: The authors described that Synthesis of new biscoumarin derivatives, in vitro cholinester-2 ase inhibition, molecular modelling and antiproliferative effect 3 in A549 human lung carcinoma cells.
This manuscript involves in in vitro assay, several cell assay and computational modeling studies.
I think, even though all studies were conducted appropriately, the enzyme assay and cell assay showed unreasonable results.

In detail, 12e shows no inhibitory activity against both hAChE and hBChE.
In rathional reasons, the authors should explain 12d along with the potency of hAChE and hBChE.
However, the authors implemented the assay Intracellular localization and mitotic activity of A549 cells with 12e, not 12d.

Author response 1: We thank the reviewer for this important point. Cytostatic/antiproliferative effects of all studied compounds 12a-12e (including 12d) were analyzed and quantified in sections 2.3.2 and 2.3.3. It was concluded that the compounds 12d and 12e showed the most potent effect on mitotic activity/cell cycle arrest. However, only the compound 12e showed decreased cell viability, indicating cytotoxic effect (Fig. 5B), worth of further analyses (mitotic activity and induction of apoptosis). In another experiments, we also performed the analysis of intracellular localization of all studied compounds 12a-12e (including 12d) on A549 cells using confocal microscopy. However, only the compound 12e showed increased fluorescence of cells indicating the accumulation of the fluorescent chemical derivative, which could be visualized and localized in cells. Therefore, in subsequent analyses of intracellular localization, mitotic activity and induction of apoptosis, we decided to focus only on the compound 12e. This is stated in manuscript (lines 538 - 563). The compound 12a was used only for comparison, which could lead to some misunderstanding. However, if it is necessary, we are ready to perform all analyses also on compound 12d, but in this case we will need significantly more time for revision (approx. 14 days). According to our opinion, the effect of the added analyses on the improvement of manuscript could be questionable, since all of the analyses (except the mitotic activity) should provide negative results.

Poin 2: In contrast, the modeling studies was carried out using 12d.

If the authors would like to claim the good potency of 12e on cell studies according to good permeability, the authors should confirm more high concentration result on enzyme assay, first.
After that, the differency of enzyme and cell results should be discussed.

 Author response 2: It is clear that derivative 12d is a potent inhibitor of hAChE, while 12e not. Therefore it is reasonable to perform the docking studies only for derivative 12d. The IC50 for derivative 12e of  > 500 µM means that due to solubility reason we could not test it in higher concentration. In fact, no effect at such high concentration (500 µM) means that there is definitely no clinical relevance by the AChE inhibition caused by this compound and therefore no rationale for targeting the Alzheimer´s disease. On the other hand, good BBB permeability may be important in targeting other CNS disorders like schizophrenia; and intracellular localization and mitotic activity is suitable for targeting cancer.

We have modified the Abstract and Conclusion accordingly

Poin 3: My request is to see higher concentration results of compound 12e on enzyme assay, at least.
I cannot decide the possibility to publicate this manusctipt without seeing correlation with the result of enzyme assay and cell assay.

Author response 3: We would like to apologize for the misunderstanding. Apparently there was some misunderstanding here, because it was not our intention to prove that the corresponding bis-coumarin derivatives would act as cholinesterase inhibitors and at the same time as anti-cancer drugs. Whereas patients with certain central nervous system (CNS) disorders are known to develop less cancer than expected, than this result that derivative 12e does not inhibit cholinesterases but on the other hand shows good antiproliferative activity is pharmacologically good because there is no reason for a potential cancer drug to inhibit AChE, as this is not necessary for this activity. Basic science increasingly indicates shared molecular mechanisms between cancer and AD and gives weight to key relevant biological theories.

Poin 1: The authors described that Synthesis of new biscoumarin derivatives, in vitro cholinester-2 ase inhibition, molecular modelling and antiproliferative effect 3 in A549 human lung carcinoma cells.
This manuscript involves in in vitro assay, several cell assay and computational modeling studies.
I think, even though all studies were conducted appropriately, the enzyme assay and cell assay showed unreasonable results.

In detail, 12e shows no inhibitory activity against both hAChE and hBChE.
In rathional reasons, the authors should explain 12d along with the potency of hAChE and hBChE.
However, the authors implemented the assay Intracellular localization and mitotic activity of A549 cells with 12e, not 12d.

Author response 1: We thank the reviewer for this important point. Cytostatic/antiproliferative effects of all studied compounds 12a-12e (including 12d) were analyzed and quantified in sections 2.3.2 and 2.3.3. It was concluded that the compounds 12d and 12e showed the most potent effect on mitotic activity/cell cycle arrest. However, only the compound 12e showed decreased cell viability, indicating cytotoxic effect (Fig. 5B), worth of further analyses (mitotic activity and induction of apoptosis). In another experiments, we also performed the analysis of intracellular localization of all studied compounds 12a-12e (including 12d) on A549 cells using confocal microscopy. However, only the compound 12e showed increased fluorescence of cells indicating the accumulation of the fluorescent chemical derivative, which could be visualized and localized in cells. Therefore, in subsequent analyses of intracellular localization, mitotic activity and induction of apoptosis, we decided to focus only on the compound 12e. This is stated in manuscript (lines 538 - 563). The compound 12a was used only for comparison, which could lead to some misunderstanding. However, if it is necessary, we are ready to perform all analyses also on compound 12d, but in this case we will need significantly more time for revision (approx. 14 days). According to our opinion, the effect of the added analyses on the improvement of manuscript could be questionable, since all of the analyses (except the mitotic activity) should provide negative results.

Poin 2: In contrast, the modeling studies was carried out using 12d.

If the authors would like to claim the good potency of 12e on cell studies according to good permeability, the authors should confirm more high concentration result on enzyme assay, first.
After that, the differency of enzyme and cell results should be discussed.

 Author response 2: It is clear that derivative 12d is a potent inhibitor of hAChE, while 12e not. Therefore it is reasonable to perform the docking studies only for derivative 12d. The IC50 for derivative 12e of  > 500 µM means that due to solubility reason we could not test it in higher concentration. In fact, no effect at such high concentration (500 µM) means that there is definitely no clinical relevance by the AChE inhibition caused by this compound and therefore no rationale for targeting the Alzheimer´s disease. On the other hand, good BBB permeability may be important in targeting other CNS disorders like schizophrenia; and intracellular localization and mitotic activity is suitable for targeting cancer.

We have modified the Abstract and Conclusion accordingly

Poin 3: My request is to see higher concentration results of compound 12e on enzyme assay, at least.
I cannot decide the possibility to publicate this manusctipt without seeing correlation with the result of enzyme assay and cell assay.

Author response 3: We would like to apologize for the misunderstanding. Apparently there was some misunderstanding here, because it was not our intention to prove that the corresponding bis-coumarin derivatives would act as cholinesterase inhibitors and at the same time as anti-cancer drugs. Whereas patients with certain central nervous system (CNS) disorders are known to develop less cancer than expected, than this result that derivative 12e does not inhibit cholinesterases but on the other hand shows good antiproliferative activity is pharmacologically good because there is no reason for a potential cancer drug to inhibit AChE, as this is not necessary for this activity. Basic science increasingly indicates shared molecular mechanisms between cancer and AD and gives weight to key relevant biological theories.

Poin 1: The authors described that Synthesis of new biscoumarin derivatives, in vitro cholinester-2 ase inhibition, molecular modelling and antiproliferative effect 3 in A549 human lung carcinoma cells.
This manuscript involves in in vitro assay, several cell assay and computational modeling studies.
I think, even though all studies were conducted appropriately, the enzyme assay and cell assay showed unreasonable results.

In detail, 12e shows no inhibitory activity against both hAChE and hBChE.
In rathional reasons, the authors should explain 12d along with the potency of hAChE and hBChE.
However, the authors implemented the assay Intracellular localization and mitotic activity of A549 cells with 12e, not 12d.

Author response 1: We thank the reviewer for this important point. Cytostatic/antiproliferative effects of all studied compounds 12a-12e (including 12d) were analyzed and quantified in sections 2.3.2 and 2.3.3. It was concluded that the compounds 12d and 12e showed the most potent effect on mitotic activity/cell cycle arrest. However, only the compound 12e showed decreased cell viability, indicating cytotoxic effect (Fig. 5B), worth of further analyses (mitotic activity and induction of apoptosis). In another experiments, we also performed the analysis of intracellular localization of all studied compounds 12a-12e (including 12d) on A549 cells using confocal microscopy. However, only the compound 12e showed increased fluorescence of cells indicating the accumulation of the fluorescent chemical derivative, which could be visualized and localized in cells. Therefore, in subsequent analyses of intracellular localization, mitotic activity and induction of apoptosis, we decided to focus only on the compound 12e. This is stated in manuscript (lines 538 - 563). The compound 12a was used only for comparison, which could lead to some misunderstanding. However, if it is necessary, we are ready to perform all analyses also on compound 12d, but in this case we will need significantly more time for revision (approx. 14 days). According to our opinion, the effect of the added analyses on the improvement of manuscript could be questionable, since all of the analyses (except the mitotic activity) should provide negative results.

Poin 2: In contrast, the modeling studies was carried out using 12d.

If the authors would like to claim the good potency of 12e on cell studies according to good permeability, the authors should confirm more high concentration result on enzyme assay, first.
After that, the differency of enzyme and cell results should be discussed.

 Author response 2: It is clear that derivative 12d is a potent inhibitor of hAChE, while 12e not. Therefore it is reasonable to perform the docking studies only for derivative 12d. The IC50 for derivative 12e of  > 500 µM means that due to solubility reason we could not test it in higher concentration. In fact, no effect at such high concentration (500 µM) means that there is definitely no clinical relevance by the AChE inhibition caused by this compound and therefore no rationale for targeting the Alzheimer´s disease. On the other hand, good BBB permeability may be important in targeting other CNS disorders like schizophrenia; and intracellular localization and mitotic activity is suitable for targeting cancer.

We have modified the Abstract and Conclusion accordingly

Poin 3: My request is to see higher concentration results of compound 12e on enzyme assay, at least.
I cannot decide the possibility to publicate this manusctipt without seeing correlation with the result of enzyme assay and cell assay.

Author response 3: We would like to apologize for the misunderstanding. Apparently there was some misunderstanding here, because it was not our intention to prove that the corresponding bis-coumarin derivatives would act as cholinesterase inhibitors and at the same time as anti-cancer drugs. Whereas patients with certain central nervous system (CNS) disorders are known to develop less cancer than expected, than this result that derivative 12e does not inhibit cholinesterases but on the other hand shows good antiproliferative activity is pharmacologically good because there is no reason for a potential cancer drug to inhibit AChE, as this is not necessary for this activity. Basic science increasingly indicates shared molecular mechanisms between cancer and AD and gives weight to key relevant biological theories.

Reviewer 2 Report

A series of novel C4-C7-tethered biscoumarin derivatives (12a-e) linked through piperazine moiety was designed, synthesized, and biologically evaluated. Biscoumarin 12d was found to be the most effective inhibitor of both acetylcholinesterase and butyrylcholinesterase. Detailed molecular modelling studies compared the accommodation of ensaculin and 12d in the hAChE active site. The ability of novel compounds to cross the BBB was predicted with a positive outcome for compound 12e. The antiproliferative effects of newly synthesized biscoumarin derivatives were tested. Intracellular localization of used derivatives in A549 cells was confirmed by confocal microscopy. Derivatives 12d and 12e showed significant antiproliferative activity in A549 cancer cells without significant effect on normal CCD-18Co cells. The inhibition of hAChE/hBChE, the antiproliferative activity on cancer cells and the ability to cross the BBB suggest the high potential of biscoumarin derivatives 12d and 12e for future development as therapeutic agents in the prevention and/or treatment of Alzheimer’s disease and cancer.

Major points:

  1. The author should explain what are the benefits of developing a dual therapy drug that can treat AD and cancer? Because these diseases do not occur at the same time.
  2. The effect of using acetylcholinesterase inhibitors to treat AD and cancer is opposite. Because the activity of acetylcholinesterase is beneficial to cancer, but harmful to AD.
  3. What is the anticoagulant activity of this biscoumarin? Because coumarin is used for anticoagulant drugs and this activity will be the side effect of this biscoumarin to be used for treating AD and cancer.

Minor points:

  1. Line 31-32: “acetylcholinesterase (hAChE, IC50 = 6.30 μM) and butyrylcholinesterase (hBChE, IC50 = 49 μM).” should be “acetylcholinesterase (AChE, IC50 = 6.30 μM) and butyrylcholinesterase (BChE, IC50 = 49 μM).”.
  2. Line 34: “hAChE” should be “human recombinant AChE (hAChE)”.
  3. Line 42, 126: “hAChE/hBChE” should be “hAChE/ humaman recombinant BchE (hBChE)”.
  4. Line 42-43; 210; 562: “blood-brain barrier” should be “BBB”.
  5. Line 59-61: “Although both patologies illnesses are multifactorial primary and associated with the aging process, they affect millions of people around the world.” may be rewritten as “Although both types of diseases are multifactorial primary diseases and are related to the aging process, they affect millions of people worldwide.”.
  6. Line 75-76: “acetylcholinesterase inhibitors (AChEIs)” should be “acetylcholinesterase (AChE) inhibitors (AChEIs)”.
  7. Line 79: “peripheral aninonic sites (PAS)” should be “PAS”.
  8. Line 130: “12a-e (n = 3, 4, 6, 7, 8)” should be “12a-e (n = 3, 4, 6, 7, 8, respectively)”.
  9. Line 150: “7-MEOTA” should be “7-methoxytacrine (7-MEOTA)”.
  10. Line 151: “IC50” should be “half maximal inhibitory concentration (IC50)”.
  11. Line 211: “blood-brain barrier (BBB) is a limiting factor in CNS” should be “BBB is a limiting factor in central nervous system (CNS)”.
  12. Line 248: “- not defined” should be “- not defined. DMSO, Dimethyl sulfoxide.”
  13. Line 260: “HUVEC” should be “Human umbilical vein endothelial cell ( HUVEC)”.
  14. Line 288-296: “hAChEI/hBChEI” should be “hAChE/hBChE”.
  15. Line 298: “AChE inhibitor” should be “AChEI”.
  16. Line 374: “MTT” should be “MTT (3-(4,5-603 dimethylthiazol-2-yl)-2,5-diphenyltertrazolium bromide)”.
  17. Line 389: “with 10% FBS” should be “with 10% fetal bovine serum (FBS)”.
  18. Line 539: “anti-ChE” should be “anti-cholinesterase”.
  19. Line 550: “KH2PO4/K2HPO4 buffer” should be “phosphate buffer”.
  20. Line 552: “ATC/BTC” should be “acetylthiocholine (ATC)/butyrylthiocholine (BTC)”.
  21. Line 565: “PBS” should be “phosphate buffered saline (PBS)”.

Author Response

A series of novel C4-C7-tethered biscoumarin derivatives (12a-e) linked through piperazine moiety was designed, synthesized, and biologically evaluated. Biscoumarin 12d was found to be the most effective inhibitor of both acetylcholinesterase and butyrylcholinesterase. Detailed molecular modelling studies compared the accommodation of ensaculin and 12d in the hAChE active site. The ability of novel compounds to cross the BBB was predicted with a positive outcome for compound 12e. The antiproliferative effects of newly synthesized biscoumarin derivatives were tested. Intracellular localization of used derivatives in A549 cells was confirmed by confocal microscopy. Derivatives 12d and 12e showed significant antiproliferative activity in A549 cancer cells without significant effect on normal CCD-18Co cells. The inhibition of hAChE/hBChE, the antiproliferative activity on cancer cells and the ability to cross the BBB suggest the high potential of biscoumarin derivatives 12d and 12e for future development as therapeutic agents in the prevention and/or treatment of Alzheimer’s disease and cancer.

 Major points:

Poin 1: The author should explain what are the benefits of developing a dual therapy drug that can treat AD and cancer? Because these diseases do not occur at the same time.

Poin 2: The effect of using acetylcholinesterase inhibitors to treat AD and cancer is opposite. Because the activity of acetylcholinesterase is beneficial to cancer, but harmful to AD.

Author response to points 1 and 2: We thank the reviewer for this important point. The aim of our study has not been to develop dual therapy drugs for AD and cancer simultaneously. Therefore, we deleted the sentences in lines 113, which could lead the reader to a misunderstanding, and deleted figure 3, too. In our study, we have tested biological potential new biscoumarine derivatives as AD or cancer therapy, because these disorders are still more prevalent in the world and have a multifactorial genesis. On the other hand, for designed new AD drugs, it is good to know, how it affects the cells in general (toxic or not toxic), therefore we have tested both the antiproliferative and acetylcholinesterase activity.  It could provide more information about a tested compound at the same time.

And vice versa for a designed drug as therapeutic agents against (lung) cancer it is not beneficial then compound act as a (catalytic) inhibitor of acetylcholinesterases. Therefore primary information about possible anticholinesterase activity of potential new anticancer (lung) designed drugs is beneficial for further study of these compounds. Moreover, in the recent study was mentioned that some acetylcholine inhibitors could inhibit proliferation of cancer cell and also some could suppress colony formation of cancer cells (mentioned in line: 156 - 162). In this case, the non-classical function AChE based on the ability of AChE to bind with a range of proteins through the PAS can play an important role (in line 143-147) because AChE is involved in apoptosis, too.

Also, in the introduction, abstract, and conclusion, were MORE clearly defined main goals of our study and more clearly presented conclusion our result (Line: 44-45; 132 – 137; 198 – 205; 215 – 216; 913 – 921))

We would like to apologize for the misunderstanding.

Poin 3: What is the anticoagulant activity of this biscoumarin? Because coumarin is used for anticoagulant drugs and this activity will be the side effect of this biscoumarin to be used for treating AD and cancer.

Author response point 3: We did not test the new compounds for anticoagulant activity because this was not our goal and we are also aware of the complexity of the models for anticoagulant activity.

Minor points:

  1. Line 31-32: “acetylcholinesterase (hAChE, IC50 = 6.30 μM) and butyrylcholinesterase (hBChE, IC50 = 49 μM).” should be “acetylcholinesterase (AChE, IC50 = 6.30 μM) and butyrylcholinesterase (BChE, IC50 = 49 μM).”.
  2. Line 34: “hAChE” should be “human recombinant AChE (hAChE)”.
  3. Line 42, 126: “hAChE/hBChE” should be “hAChE/ humaman recombinant BchE (hBChE)”.
  4. Line 42-43; 210; 562: “blood-brain barrier” should be “BBB”.
  5. Line 59-61: “Although both patologies illnesses are multifactorial primary and associated with the aging process, they affect millions of people around the world.” may be rewritten as “Although both types of diseases are multifactorial primary diseases and are related to the aging process, they affect millions of people worldwide.”.
  6. Line 75-76: “acetylcholinesterase inhibitors (AChEIs)” should be “acetylcholinesterase (AChE) inhibitors (AChEIs)”.
  7. Line 79: “peripheral aninonic sites (PAS)” should be “PAS”.
  8. Line 130: “12a-(n = 3, 4, 6, 7, 8)” should be “12a-(n = 3, 4, 6, 7, 8, respectively)”.
  9. Line 150: “7-MEOTA” should be “7-methoxytacrine (7-MEOTA)”.
  10. Line 151: “IC50” should be “half maximal inhibitory concentration (IC50)”.
  11. Line 211: “blood-brain barrier (BBB) is a limiting factor in CNS” should be “BBB is a limiting factor in central nervous system (CNS)”.
  12. Line 248: “- not defined” should be “- not defined. DMSO, Dimethyl sulfoxide.”
  13. Line 260: “HUVEC” should be “Human umbilical vein endothelial cell ( HUVEC)”.
  14. Line 288-296: “hAChEI/hBChEI” should be “hAChE/hBChE”.
  15. Line 298: “AChE inhibitor” should be “AChEI”.
  16. Line 374: “MTT” should be “MTT (3-(4,5-603 dimethylthiazol-2-yl)-2,5-diphenyltertrazolium bromide)”.
  17. Line 389: “with 10% FBS” should be “with 10% fetal bovine serum (FBS)”.
  18. Line 539: “anti-ChE” should be “anti-cholinesterase”.
  19. Line 550: “KH2PO4/K2HPO4 buffer” should be “phosphate buffer”.
  20. Line 552: “ATC/BTC” should be “acetylthiocholine (ATC)/butyrylthiocholine (BTC)”.
  21. Line 565: “PBS” should be “phosphate buffered saline (PBS)”.

Author response minor points: We have accepted all referee´s recommendations and corrected our manuscript as follows:

  1. Line 31-32: “acetylcholinesterase (hAChE, IC50 = 6.30 μM) and butyrylcholinesterase (hBChE, IC50 = 49 μM).” should be “acetylcholinesterase (AChE, IC50 = 6.30 μM) and butyrylcholinesterase (BChE, IC50 = 49 μM).”.
  2. Line 34: “hAChE” should be “human recombinant AChE (hAChE)”.
  3. Line 42, 126: “hAChE/hBChE” should be “hAChE/ humaman recombinant BchE (hBChE)”.
  4. Line 42-43; 210; 562: “blood-brain barrier” should be “BBB”.
  5. Line 59-61: “Although both patologies illnesses are multifactorial primary and associated with the aging process, they affect millions of people around the world.” may be rewritten as “Although both types of diseases are multifactorial primary diseases and are related to the aging process, they affect millions of people worldwide.”.
  6. Line 75-76: “acetylcholinesterase inhibitors (AChEIs)” should be “acetylcholinesterase (AChE) inhibitors (AChEIs)”.
  7. Line 79: “peripheral aninonic sites (PAS)” should be “PAS”.
  8. Line 130: “12a-(n = 3, 4, 6, 7, 8)” should be “12a-(n = 3, 4, 6, 7, 8, respectively)”.
  9. Line 150: “7-MEOTA” should be “7-methoxytacrine (7-MEOTA)”.
  10. Line 151: “IC50” should be “half maximal inhibitory concentration (IC50)”.
  11. Line 211: “blood-brain barrier (BBB) is a limiting factor in CNS” should be “BBB is a limiting factor in central nervous system (CNS)”.
  12. Line 248: “- not defined” should be “- not defined. DMSO, Dimethyl sulfoxide.”
  13. Line 260: “HUVEC” should be “Human umbilical vein endothelial cell ( HUVEC)”.
  14. Line 288-296: “hAChEI/hBChEI” should be “hAChE/hBChE”.
  15. Line 298: “AChE inhibitor” should be “AChEI”.
  16. Line 374: “MTT” should be “MTT (3-(4,5-603 dimethylthiazol-2-yl)-2,5-diphenyltertrazolium bromide)”.
  17. Line 389: “with 10% FBS” should be “with 10% fetal bovine serum (FBS)”.
  18. Line 539: “anti-ChE” should be “anti-cholinesterase”.
  19. Line 550: “KH2PO4/K2HPO4 buffer” should be “phosphate buffer”.
  20. Line 552: “ATC/BTC” should be “acetylthiocholine (ATC)/butyrylthiocholine (BTC)”.
  21. Line 565: “PBS” should be “phosphate buffered saline (PBS)”.

    A series of novel C4-C7-tethered biscoumarin derivatives (12a-e) linked through piperazine moiety was designed, synthesized, and biologically evaluated. Biscoumarin 12d was found to be the most effective inhibitor of both acetylcholinesterase and butyrylcholinesterase. Detailed molecular modelling studies compared the accommodation of ensaculin and 12d in the hAChE active site. The ability of novel compounds to cross the BBB was predicted with a positive outcome for compound 12e. The antiproliferative effects of newly synthesized biscoumarin derivatives were tested. Intracellular localization of used derivatives in A549 cells was confirmed by confocal microscopy. Derivatives 12d and 12e showed significant antiproliferative activity in A549 cancer cells without significant effect on normal CCD-18Co cells. The inhibition of hAChE/hBChE, the antiproliferative activity on cancer cells and the ability to cross the BBB suggest the high potential of biscoumarin derivatives 12d and 12e for future development as therapeutic agents in the prevention and/or treatment of Alzheimer’s disease and cancer.

     Major points:

    Poin 1: The author should explain what are the benefits of developing a dual therapy drug that can treat AD and cancer? Because these diseases do not occur at the same time.

    Poin 2: The effect of using acetylcholinesterase inhibitors to treat AD and cancer is opposite. Because the activity of acetylcholinesterase is beneficial to cancer, but harmful to AD.

    Author response to points 1 and 2: We thank the reviewer for this important point. The aim of our study has not been to develop dual therapy drugs for AD and cancer simultaneously. Therefore, we deleted the sentences in lines 113, which could lead the reader to a misunderstanding, and deleted figure 3, too. In our study, we have tested biological potential new biscoumarine derivatives as AD or cancer therapy, because these disorders are still more prevalent in the world and have a multifactorial genesis. On the other hand, for designed new AD drugs, it is good to know, how it affects the cells in general (toxic or not toxic), therefore we have tested both the antiproliferative and acetylcholinesterase activity.  It could provide more information about a tested compound at the same time.

    And vice versa for a designed drug as therapeutic agents against (lung) cancer it is not beneficial then compound act as a (catalytic) inhibitor of acetylcholinesterases. Therefore primary information about possible anticholinesterase activity of potential new anticancer (lung) designed drugs is beneficial for further study of these compounds. Moreover, in the recent study was mentioned that some acetylcholine inhibitors could inhibit proliferation of cancer cell and also some could suppress colony formation of cancer cells (mentioned in line: 156 - 162). In this case, the non-classical function AChE based on the ability of AChE to bind with a range of proteins through the PAS can play an important role (in line 143-147) because AChE is involved in apoptosis, too.

    Also, in the introduction, abstract, and conclusion, were MORE clearly defined main goals of our study and more clearly presented conclusion our result (Line: 44-45; 132 – 137; 198 – 205; 215 – 216; 913 – 921))

    We would like to apologize for the misunderstanding.

    Poin 3: What is the anticoagulant activity of this biscoumarin? Because coumarin is used for anticoagulant drugs and this activity will be the side effect of this biscoumarin to be used for treating AD and cancer.

    Author response point 3: We did not test the new compounds for anticoagulant activity because this was not our goal and we are also aware of the complexity of the models for anticoagulant activity.

    Minor points:

    1. Line 31-32: “acetylcholinesterase (hAChE, IC50 = 6.30 μM) and butyrylcholinesterase (hBChE, IC50 = 49 μM).” should be “acetylcholinesterase (AChE, IC50 = 6.30 μM) and butyrylcholinesterase (BChE, IC50 = 49 μM).”.
    2. Line 34: “hAChE” should be “human recombinant AChE (hAChE)”.
    3. Line 42, 126: “hAChE/hBChE” should be “hAChE/ humaman recombinant BchE (hBChE)”.
    4. Line 42-43; 210; 562: “blood-brain barrier” should be “BBB”.
    5. Line 59-61: “Although both patologies illnesses are multifactorial primary and associated with the aging process, they affect millions of people around the world.” may be rewritten as “Although both types of diseases are multifactorial primary diseases and are related to the aging process, they affect millions of people worldwide.”.
    6. Line 75-76: “acetylcholinesterase inhibitors (AChEIs)” should be “acetylcholinesterase (AChE) inhibitors (AChEIs)”.
    7. Line 79: “peripheral aninonic sites (PAS)” should be “PAS”.
    8. Line 130: “12a-(n = 3, 4, 6, 7, 8)” should be “12a-(n = 3, 4, 6, 7, 8, respectively)”.
    9. Line 150: “7-MEOTA” should be “7-methoxytacrine (7-MEOTA)”.
    10. Line 151: “IC50” should be “half maximal inhibitory concentration (IC50)”.
    11. Line 211: “blood-brain barrier (BBB) is a limiting factor in CNS” should be “BBB is a limiting factor in central nervous system (CNS)”.
    12. Line 248: “- not defined” should be “- not defined. DMSO, Dimethyl sulfoxide.”
    13. Line 260: “HUVEC” should be “Human umbilical vein endothelial cell ( HUVEC)”.
    14. Line 288-296: “hAChEI/hBChEI” should be “hAChE/hBChE”.
    15. Line 298: “AChE inhibitor” should be “AChEI”.
    16. Line 374: “MTT” should be “MTT (3-(4,5-603 dimethylthiazol-2-yl)-2,5-diphenyltertrazolium bromide)”.
    17. Line 389: “with 10% FBS” should be “with 10% fetal bovine serum (FBS)”.
    18. Line 539: “anti-ChE” should be “anti-cholinesterase”.
    19. Line 550: “KH2PO4/K2HPO4 buffer” should be “phosphate buffer”.
    20. Line 552: “ATC/BTC” should be “acetylthiocholine (ATC)/butyrylthiocholine (BTC)”.
    21. Line 565: “PBS” should be “phosphate buffered saline (PBS)”.

    Author response minor points: We have accepted all referee´s recommendations and corrected our manuscript as follows:

    1. Line 31-32: “acetylcholinesterase (hAChE, IC50 = 6.30 μM) and butyrylcholinesterase (hBChE, IC50 = 49 μM).” should be “acetylcholinesterase (AChE, IC50 = 6.30 μM) and butyrylcholinesterase (BChE, IC50 = 49 μM).”.
    2. Line 34: “hAChE” should be “human recombinant AChE (hAChE)”.
    3. Line 42, 126: “hAChE/hBChE” should be “hAChE/ humaman recombinant BchE (hBChE)”.
    4. Line 42-43; 210; 562: “blood-brain barrier” should be “BBB”.
    5. Line 59-61: “Although both patologies illnesses are multifactorial primary and associated with the aging process, they affect millions of people around the world.” may be rewritten as “Although both types of diseases are multifactorial primary diseases and are related to the aging process, they affect millions of people worldwide.”.
    6. Line 75-76: “acetylcholinesterase inhibitors (AChEIs)” should be “acetylcholinesterase (AChE) inhibitors (AChEIs)”.
    7. Line 79: “peripheral aninonic sites (PAS)” should be “PAS”.
    8. Line 130: “12a-(n = 3, 4, 6, 7, 8)” should be “12a-(n = 3, 4, 6, 7, 8, respectively)”.
    9. Line 150: “7-MEOTA” should be “7-methoxytacrine (7-MEOTA)”.
    10. Line 151: “IC50” should be “half maximal inhibitory concentration (IC50)”.
    11. Line 211: “blood-brain barrier (BBB) is a limiting factor in CNS” should be “BBB is a limiting factor in central nervous system (CNS)”.
    12. Line 248: “- not defined” should be “- not defined. DMSO, Dimethyl sulfoxide.”
    13. Line 260: “HUVEC” should be “Human umbilical vein endothelial cell ( HUVEC)”.
    14. Line 288-296: “hAChEI/hBChEI” should be “hAChE/hBChE”.
    15. Line 298: “AChE inhibitor” should be “AChEI”.
    16. Line 374: “MTT” should be “MTT (3-(4,5-603 dimethylthiazol-2-yl)-2,5-diphenyltertrazolium bromide)”.
    17. Line 389: “with 10% FBS” should be “with 10% fetal bovine serum (FBS)”.
    18. Line 539: “anti-ChE” should be “anti-cholinesterase”.
    19. Line 550: “KH2PO4/K2HPO4 buffer” should be “phosphate buffer”.
    20. Line 552: “ATC/BTC” should be “acetylthiocholine (ATC)/butyrylthiocholine (BTC)”.
    21. Line 565: “PBS” should be “phosphate buffered saline (PBS)”.

      A series of novel C4-C7-tethered biscoumarin derivatives (12a-e) linked through piperazine moiety was designed, synthesized, and biologically evaluated. Biscoumarin 12d was found to be the most effective inhibitor of both acetylcholinesterase and butyrylcholinesterase. Detailed molecular modelling studies compared the accommodation of ensaculin and 12d in the hAChE active site. The ability of novel compounds to cross the BBB was predicted with a positive outcome for compound 12e. The antiproliferative effects of newly synthesized biscoumarin derivatives were tested. Intracellular localization of used derivatives in A549 cells was confirmed by confocal microscopy. Derivatives 12d and 12e showed significant antiproliferative activity in A549 cancer cells without significant effect on normal CCD-18Co cells. The inhibition of hAChE/hBChE, the antiproliferative activity on cancer cells and the ability to cross the BBB suggest the high potential of biscoumarin derivatives 12d and 12e for future development as therapeutic agents in the prevention and/or treatment of Alzheimer’s disease and cancer.

       Major points:

      Poin 1: The author should explain what are the benefits of developing a dual therapy drug that can treat AD and cancer? Because these diseases do not occur at the same time.

      Poin 2: The effect of using acetylcholinesterase inhibitors to treat AD and cancer is opposite. Because the activity of acetylcholinesterase is beneficial to cancer, but harmful to AD.

      Author response to points 1 and 2: We thank the reviewer for this important point. The aim of our study has not been to develop dual therapy drugs for AD and cancer simultaneously. Therefore, we deleted the sentences in lines 113, which could lead the reader to a misunderstanding, and deleted figure 3, too. In our study, we have tested biological potential new biscoumarine derivatives as AD or cancer therapy, because these disorders are still more prevalent in the world and have a multifactorial genesis. On the other hand, for designed new AD drugs, it is good to know, how it affects the cells in general (toxic or not toxic), therefore we have tested both the antiproliferative and acetylcholinesterase activity.  It could provide more information about a tested compound at the same time.

      And vice versa for a designed drug as therapeutic agents against (lung) cancer it is not beneficial then compound act as a (catalytic) inhibitor of acetylcholinesterases. Therefore primary information about possible anticholinesterase activity of potential new anticancer (lung) designed drugs is beneficial for further study of these compounds. Moreover, in the recent study was mentioned that some acetylcholine inhibitors could inhibit proliferation of cancer cell and also some could suppress colony formation of cancer cells (mentioned in line: 156 - 162). In this case, the non-classical function AChE based on the ability of AChE to bind with a range of proteins through the PAS can play an important role (in line 143-147) because AChE is involved in apoptosis, too.

      Also, in the introduction, abstract, and conclusion, were MORE clearly defined main goals of our study and more clearly presented conclusion our result (Line: 44-45; 132 – 137; 198 – 205; 215 – 216; 913 – 921))

      We would like to apologize for the misunderstanding.

      Poin 3: What is the anticoagulant activity of this biscoumarin? Because coumarin is used for anticoagulant drugs and this activity will be the side effect of this biscoumarin to be used for treating AD and cancer.

      Author response point 3: We did not test the new compounds for anticoagulant activity because this was not our goal and we are also aware of the complexity of the models for anticoagulant activity.

      Minor points:

      1. Line 31-32: “acetylcholinesterase (hAChE, IC50 = 6.30 μM) and butyrylcholinesterase (hBChE, IC50 = 49 μM).” should be “acetylcholinesterase (AChE, IC50 = 6.30 μM) and butyrylcholinesterase (BChE, IC50 = 49 μM).”.
      2. Line 34: “hAChE” should be “human recombinant AChE (hAChE)”.
      3. Line 42, 126: “hAChE/hBChE” should be “hAChE/ humaman recombinant BchE (hBChE)”.
      4. Line 42-43; 210; 562: “blood-brain barrier” should be “BBB”.
      5. Line 59-61: “Although both patologies illnesses are multifactorial primary and associated with the aging process, they affect millions of people around the world.” may be rewritten as “Although both types of diseases are multifactorial primary diseases and are related to the aging process, they affect millions of people worldwide.”.
      6. Line 75-76: “acetylcholinesterase inhibitors (AChEIs)” should be “acetylcholinesterase (AChE) inhibitors (AChEIs)”.
      7. Line 79: “peripheral aninonic sites (PAS)” should be “PAS”.
      8. Line 130: “12a-(n = 3, 4, 6, 7, 8)” should be “12a-(n = 3, 4, 6, 7, 8, respectively)”.
      9. Line 150: “7-MEOTA” should be “7-methoxytacrine (7-MEOTA)”.
      10. Line 151: “IC50” should be “half maximal inhibitory concentration (IC50)”.
      11. Line 211: “blood-brain barrier (BBB) is a limiting factor in CNS” should be “BBB is a limiting factor in central nervous system (CNS)”.
      12. Line 248: “- not defined” should be “- not defined. DMSO, Dimethyl sulfoxide.”
      13. Line 260: “HUVEC” should be “Human umbilical vein endothelial cell ( HUVEC)”.
      14. Line 288-296: “hAChEI/hBChEI” should be “hAChE/hBChE”.
      15. Line 298: “AChE inhibitor” should be “AChEI”.
      16. Line 374: “MTT” should be “MTT (3-(4,5-603 dimethylthiazol-2-yl)-2,5-diphenyltertrazolium bromide)”.
      17. Line 389: “with 10% FBS” should be “with 10% fetal bovine serum (FBS)”.
      18. Line 539: “anti-ChE” should be “anti-cholinesterase”.
      19. Line 550: “KH2PO4/K2HPO4 buffer” should be “phosphate buffer”.
      20. Line 552: “ATC/BTC” should be “acetylthiocholine (ATC)/butyrylthiocholine (BTC)”.
      21. Line 565: “PBS” should be “phosphate buffered saline (PBS)”.

      Author response minor points: We have accepted all referee´s recommendations and corrected our manuscript as follows:

      1. Line 31-32: “acetylcholinesterase (hAChE, IC50 = 6.30 μM) and butyrylcholinesterase (hBChE, IC50 = 49 μM).” should be “acetylcholinesterase (AChE, IC50 = 6.30 μM) and butyrylcholinesterase (BChE, IC50 = 49 μM).”.
      2. Line 34: “hAChE” should be “human recombinant AChE (hAChE)”.
      3. Line 42, 126: “hAChE/hBChE” should be “hAChE/ humaman recombinant BchE (hBChE)”.
      4. Line 42-43; 210; 562: “blood-brain barrier” should be “BBB”.
      5. Line 59-61: “Although both patologies illnesses are multifactorial primary and associated with the aging process, they affect millions of people around the world.” may be rewritten as “Although both types of diseases are multifactorial primary diseases and are related to the aging process, they affect millions of people worldwide.”.
      6. Line 75-76: “acetylcholinesterase inhibitors (AChEIs)” should be “acetylcholinesterase (AChE) inhibitors (AChEIs)”.
      7. Line 79: “peripheral aninonic sites (PAS)” should be “PAS”.
      8. Line 130: “12a-(n = 3, 4, 6, 7, 8)” should be “12a-(n = 3, 4, 6, 7, 8, respectively)”.
      9. Line 150: “7-MEOTA” should be “7-methoxytacrine (7-MEOTA)”.
      10. Line 151: “IC50” should be “half maximal inhibitory concentration (IC50)”.
      11. Line 211: “blood-brain barrier (BBB) is a limiting factor in CNS” should be “BBB is a limiting factor in central nervous system (CNS)”.
      12. Line 248: “- not defined” should be “- not defined. DMSO, Dimethyl sulfoxide.”
      13. Line 260: “HUVEC” should be “Human umbilical vein endothelial cell ( HUVEC)”.
      14. Line 288-296: “hAChEI/hBChEI” should be “hAChE/hBChE”.
      15. Line 298: “AChE inhibitor” should be “AChEI”.
      16. Line 374: “MTT” should be “MTT (3-(4,5-603 dimethylthiazol-2-yl)-2,5-diphenyltertrazolium bromide)”.
      17. Line 389: “with 10% FBS” should be “with 10% fetal bovine serum (FBS)”.
      18. Line 539: “anti-ChE” should be “anti-cholinesterase”.
      19. Line 550: “KH2PO4/K2HPO4 buffer” should be “phosphate buffer”.
      20. Line 552: “ATC/BTC” should be “acetylthiocholine (ATC)/butyrylthiocholine (BTC)”.
      21. Line 565: “PBS” should be “phosphate buffered saline (PBS)”.

        A series of novel C4-C7-tethered biscoumarin derivatives (12a-e) linked through piperazine moiety was designed, synthesized, and biologically evaluated. Biscoumarin 12d was found to be the most effective inhibitor of both acetylcholinesterase and butyrylcholinesterase. Detailed molecular modelling studies compared the accommodation of ensaculin and 12d in the hAChE active site. The ability of novel compounds to cross the BBB was predicted with a positive outcome for compound 12e. The antiproliferative effects of newly synthesized biscoumarin derivatives were tested. Intracellular localization of used derivatives in A549 cells was confirmed by confocal microscopy. Derivatives 12d and 12e showed significant antiproliferative activity in A549 cancer cells without significant effect on normal CCD-18Co cells. The inhibition of hAChE/hBChE, the antiproliferative activity on cancer cells and the ability to cross the BBB suggest the high potential of biscoumarin derivatives 12d and 12e for future development as therapeutic agents in the prevention and/or treatment of Alzheimer’s disease and cancer.

         Major points:

        Poin 1: The author should explain what are the benefits of developing a dual therapy drug that can treat AD and cancer? Because these diseases do not occur at the same time.

        Poin 2: The effect of using acetylcholinesterase inhibitors to treat AD and cancer is opposite. Because the activity of acetylcholinesterase is beneficial to cancer, but harmful to AD.

        Author response to points 1 and 2: We thank the reviewer for this important point. The aim of our study has not been to develop dual therapy drugs for AD and cancer simultaneously. Therefore, we deleted the sentences in lines 113, which could lead the reader to a misunderstanding, and deleted figure 3, too. In our study, we have tested biological potential new biscoumarine derivatives as AD or cancer therapy, because these disorders are still more prevalent in the world and have a multifactorial genesis. On the other hand, for designed new AD drugs, it is good to know, how it affects the cells in general (toxic or not toxic), therefore we have tested both the antiproliferative and acetylcholinesterase activity.  It could provide more information about a tested compound at the same time.

        And vice versa for a designed drug as therapeutic agents against (lung) cancer it is not beneficial then compound act as a (catalytic) inhibitor of acetylcholinesterases. Therefore primary information about possible anticholinesterase activity of potential new anticancer (lung) designed drugs is beneficial for further study of these compounds. Moreover, in the recent study was mentioned that some acetylcholine inhibitors could inhibit proliferation of cancer cell and also some could suppress colony formation of cancer cells (mentioned in line: 156 - 162). In this case, the non-classical function AChE based on the ability of AChE to bind with a range of proteins through the PAS can play an important role (in line 143-147) because AChE is involved in apoptosis, too.

        Also, in the introduction, abstract, and conclusion, were MORE clearly defined main goals of our study and more clearly presented conclusion our result (Line: 44-45; 132 – 137; 198 – 205; 215 – 216; 913 – 921))

        We would like to apologize for the misunderstanding.

        Poin 3: What is the anticoagulant activity of this biscoumarin? Because coumarin is used for anticoagulant drugs and this activity will be the side effect of this biscoumarin to be used for treating AD and cancer.

        Author response point 3: We did not test the new compounds for anticoagulant activity because this was not our goal and we are also aware of the complexity of the models for anticoagulant activity.

        Minor points:

        1. Line 31-32: “acetylcholinesterase (hAChE, IC50 = 6.30 μM) and butyrylcholinesterase (hBChE, IC50 = 49 μM).” should be “acetylcholinesterase (AChE, IC50 = 6.30 μM) and butyrylcholinesterase (BChE, IC50 = 49 μM).”.
        2. Line 34: “hAChE” should be “human recombinant AChE (hAChE)”.
        3. Line 42, 126: “hAChE/hBChE” should be “hAChE/ humaman recombinant BchE (hBChE)”.
        4. Line 42-43; 210; 562: “blood-brain barrier” should be “BBB”.
        5. Line 59-61: “Although both patologies illnesses are multifactorial primary and associated with the aging process, they affect millions of people around the world.” may be rewritten as “Although both types of diseases are multifactorial primary diseases and are related to the aging process, they affect millions of people worldwide.”.
        6. Line 75-76: “acetylcholinesterase inhibitors (AChEIs)” should be “acetylcholinesterase (AChE) inhibitors (AChEIs)”.
        7. Line 79: “peripheral aninonic sites (PAS)” should be “PAS”.
        8. Line 130: “12a-(n = 3, 4, 6, 7, 8)” should be “12a-(n = 3, 4, 6, 7, 8, respectively)”.
        9. Line 150: “7-MEOTA” should be “7-methoxytacrine (7-MEOTA)”.
        10. Line 151: “IC50” should be “half maximal inhibitory concentration (IC50)”.
        11. Line 211: “blood-brain barrier (BBB) is a limiting factor in CNS” should be “BBB is a limiting factor in central nervous system (CNS)”.
        12. Line 248: “- not defined” should be “- not defined. DMSO, Dimethyl sulfoxide.”
        13. Line 260: “HUVEC” should be “Human umbilical vein endothelial cell ( HUVEC)”.
        14. Line 288-296: “hAChEI/hBChEI” should be “hAChE/hBChE”.
        15. Line 298: “AChE inhibitor” should be “AChEI”.
        16. Line 374: “MTT” should be “MTT (3-(4,5-603 dimethylthiazol-2-yl)-2,5-diphenyltertrazolium bromide)”.
        17. Line 389: “with 10% FBS” should be “with 10% fetal bovine serum (FBS)”.
        18. Line 539: “anti-ChE” should be “anti-cholinesterase”.
        19. Line 550: “KH2PO4/K2HPO4 buffer” should be “phosphate buffer”.
        20. Line 552: “ATC/BTC” should be “acetylthiocholine (ATC)/butyrylthiocholine (BTC)”.
        21. Line 565: “PBS” should be “phosphate buffered saline (PBS)”.

        Author response minor points: We have accepted all referee´s recommendations and corrected our manuscript as follows:

        1. Line 31-32: “acetylcholinesterase (hAChE, IC50 = 6.30 μM) and butyrylcholinesterase (hBChE, IC50 = 49 μM).” should be “acetylcholinesterase (AChE, IC50 = 6.30 μM) and butyrylcholinesterase (BChE, IC50 = 49 μM).”.
        2. Line 34: “hAChE” should be “human recombinant AChE (hAChE)”.
        3. Line 42, 126: “hAChE/hBChE” should be “hAChE/ humaman recombinant BchE (hBChE)”.
        4. Line 42-43; 210; 562: “blood-brain barrier” should be “BBB”.
        5. Line 59-61: “Although both patologies illnesses are multifactorial primary and associated with the aging process, they affect millions of people around the world.” may be rewritten as “Although both types of diseases are multifactorial primary diseases and are related to the aging process, they affect millions of people worldwide.”.
        6. Line 75-76: “acetylcholinesterase inhibitors (AChEIs)” should be “acetylcholinesterase (AChE) inhibitors (AChEIs)”.
        7. Line 79: “peripheral aninonic sites (PAS)” should be “PAS”.
        8. Line 130: “12a-(n = 3, 4, 6, 7, 8)” should be “12a-(n = 3, 4, 6, 7, 8, respectively)”.
        9. Line 150: “7-MEOTA” should be “7-methoxytacrine (7-MEOTA)”.
        10. Line 151: “IC50” should be “half maximal inhibitory concentration (IC50)”.
        11. Line 211: “blood-brain barrier (BBB) is a limiting factor in CNS” should be “BBB is a limiting factor in central nervous system (CNS)”.
        12. Line 248: “- not defined” should be “- not defined. DMSO, Dimethyl sulfoxide.”
        13. Line 260: “HUVEC” should be “Human umbilical vein endothelial cell ( HUVEC)”.
        14. Line 288-296: “hAChEI/hBChEI” should be “hAChE/hBChE”.
        15. Line 298: “AChE inhibitor” should be “AChEI”.
        16. Line 374: “MTT” should be “MTT (3-(4,5-603 dimethylthiazol-2-yl)-2,5-diphenyltertrazolium bromide)”.
        17. Line 389: “with 10% FBS” should be “with 10% fetal bovine serum (FBS)”.
        18. Line 539: “anti-ChE” should be “anti-cholinesterase”.
        19. Line 550: “KH2PO4/K2HPO4 buffer” should be “phosphate buffer”.
        20. Line 552: “ATC/BTC” should be “acetylthiocholine (ATC)/butyrylthiocholine (BTC)”.
        21. Line 565: “PBS” should be “phosphate buffered saline (PBS)”.

Round 2

Reviewer 2 Report

The revision is complete, and I have no further comments on this manuscript.